# Mechanistic and pharmacodynamic studies of DuoBody-CD3x5T4 in preclinical tumor models

Kristel Kemper[1], Ellis Gielen[1], Peter Boross[1], Mischa Houtkamp[1], Theo S Plantinga[1], Stefanie AH de Poot[1], Saskia M Burm[1], Maarten L Janmaat[1], Louise A Koopman[1], Edward N van den Brink[1], Rik Rademaker[1], Dennis Verzijl[1], Patrick J Engelberts[1], David Satijn[1], A Kate Sasser[2], Esther CW Breij[1]

CD3 bispecific antibodies (bsAbs) show great promise as anti-cancer therapeutics. Here, we show in-depth mechanistic studies of a CD3 bsAb in solid cancer, using DuoBody-CD3x5T4. Cross-linking T cells with tumor cells expressing the oncofetal antigen 5T4 was required to induce cytotoxicity. Naive and memory CD4[+] and CD8[+] T cells were equally effective at mediating cytotoxicity, and DuoBody-CD3x5T4 induced partial differentiation of naive T-cell subsets into memory-like cells. Tumor cell kill was associated with T-cell activation, proliferation, and production of cytokines, granzyme B, and perforin. Genetic knockout of *FAS* or *IFNGR1* in 5T4[+] tumor cells abrogated tumor cell kill. In the presence of 5T4[+] tumor cells, bystander kill of 5T4[−] but not of 5T4[−]IFNGR1[−] tumor cells was observed. In humanized xenograft models, DuoBody-CD3x5T4 antitumor activity was associated with intratumoral and peripheral blood T-cell activation. Lastly, in dissociated patient-derived tumor samples, DuoBody-CD3x5T4 activated tumor-infiltrating lymphocytes and induced tumor-cell cytotoxicity, even when most tumor-infiltrating lymphocytes expressed PD-1. These data provide an in-depth view on the mechanism of action of a CD3 bsAb in preclinical models of solid cancer.

## Introduction

CD3 bispecific antibodies (bsAbs) redirect the cytotoxic activity of T cells toward tumor cells by physically linking the T-cell antigen CD3 with a tumor-associated antigen. Antibody-mediated cross-linking of the CD3/TCR complex bypasses MHC restriction and is therefore independent of the antigen specificity of the TCR. T-cell activation and subsequent T cell–mediated tumor cell kill can occur via two distinct pathways: the perforin-granzyme B pathway and the death receptor pathway. Historically, CD8[+] T cell–mediated killing is thought to occur predominantly via the perforin-granzyme B pathway, whereas CD4[+] T cell–mediated killing is thought to rely on

activation of the death receptor Fas on tumor cells by Fas ligand expressed on T cells (Shresta et al, 1998). IFNγ, which is produced by T cells in response to T-cell activation, can induce expression of Fas. IFNγ signaling has been implicated in CD3 bsAb-induced T cell–mediated killing (Arenas et al, 2021; Liu et al, 2021), not only of antigen-positive cells but also of antigen-negative cells by bystander killing, which was shown to be Fas-dependent (Ross et al, 2017; Upadhyay et al, 2021).

The concept of T-cell redirection has been successfully adopted by the CD3xCD19 bispecific T-cell engager (BiTE) blinatumomab, an approved CD3 bsAb for the treatment of acute lymphoblastic leukemia. Although a large number of CD3 bsAbs are undergoing clinical evaluation for the treatment of hematological malignancies, the number of CD3 bsAbs in clinical development for the treatment of solid tumors is limited (Dahlen et al, 2018; Velasquez et al, 2018). The oncofetal antigen 5T4, originally identified on human trophoblasts (Hole & Stern, 1988) and also referred to as trophoblast glycoprotein (TPBG), is expressed on multiple solid tumor types with reported limited expression in nonmalignant tissue (Southall et al, 1990; Ali et al, 2001; Al-Taei et al, 2012; Harper et al, 2017; Stern & Harrop, 2017; Alam et al, 2018; Schunselaar et al, 2018). In embryonic development, expression of 5T4 has been associated with epithelial-to-mesenchymal transition (EMT) (Ward et al, 2006; Eastham et al, 2007), C-X-C motif chemokine receptor 4 (CXCR4)/-ligand 12 (CXCL12)–induced chemotaxis (Southgate et al, 2010; McGinn et al, 2012; Stern et al, 2014; Puchert et al, 2018), and noncanonical Wnt signaling (Stern et al, 2014; He et al, 2015; Harrop et al, 2019). In cancer cells, 5T4 expression has also been connected to noncanonical Wnt signaling (Carsberg et al, 1996; Eastham et al, 2007; Kagermeier-Schenk et al, 2011; He et al, 2015; Harrop et al, 2019), but no association was found with the CXCR4/CXCL12 pathway (Puchert et al, 2018). Individual reports have suggested 5T4 to be a cancer stem cell marker (Damelin et al, 2011; Kerk et al, 2017), although limited follow-up studies have been published. Its expression has been associated with increased metastatic burden, worse clinical outcome, and more aggressive disease in several solid cancer indications (Starzynska et al, 1994, 1998; Wrigley et al, 1995; Mulder et al, 1997).

---

[1]Genmab, Utrecht, The Netherlands    [2]Genmab, Princeton, NJ, USA

Correspondence: ebj@genmab.com

In this report, we describe mechanistic and pharmacodynamic (PD) studies performed to obtain an in-depth preclinical understanding of CD3 bsAbs in solid cancer, using DuoBody-CD3x5T4. DuoBody-CD3x5T4 is a full-length immunoglobulin (Ig)G1 bsAb that was generated using controlled Fab-arm exchange (DuoBody Technology [Labrijn et al, 2013; Labrijn et al, 2014]) and that contains a functionally inactive Fc region and retains IgG1-like pharmacokinetic characteristics (Engelberts et al, 2020). The preclinical mechanism of action of DuoBody-CD3x5T4 and its capacity to induce antitumor activity in preclinical models were studied in vitro, in vivo, and ex vivo.

# Results

### Generation of DuoBody-CD3x5T4

Highly prevalent 5T4 expression in solid tumors (Fig S1 and Table S1) and reported limited expression in normal tissues (Southall et al, 1990; Ali et al, 2001; Alam et al, 2018) suggest that 5T4 could be a promising target for an anticancer therapeutic. DuoBody-CD3x5T4 is a bispecific IgG1 antibody with a functionally inactive Fc region and was generated from a humanized CD3ε antibody (Pessano et al, 1985) and a human 5T4 antibody through controlled Fab-arm exchange (cFAE) (Labrijn et al, 2013, 2014). DuoBody-CD3x5T4 showed high-affinity binding to recombinant 5T4 ($K_D$ = 2.9 ± 0.4 nM) (Fig S2A) and membrane-expressed 5T4 (Fig S2B). The affinity of DuoBody-CD3x5T4 for human CD3ε was 310 ± 27 nM (Fig S2C). Only limited binding of DuoBody-CD3x5T4 to human CD3ε expressed on Jurkat (acute T-cell leukemia) cells could be detected by flow cytometry, in line with the relatively low affinity for CD3ε (Fig S2D).

### DuoBody-CD3x5T4 induces T cell–mediated cytotoxicity associated with T-cell activation, proliferation, and cytokine production

DuoBody-CD3x5T4 is hypothesized to cross-link CD3ε-expressing T cells with 5T4-expressing tumor cells, thereby inducing T cell–mediated kill of 5T4-expressing tumor cells, associated with T-cell activation, T-cell proliferation, and production of inflammatory mediators and effector molecules. In cocultures of T cells and 5T4⁺ MDA-MB-468 breast cancer cells, at varying effector:target cell (E:T) ratios, DuoBody-CD3x5T4 induced dose-dependent cytotoxicity at all E:T ratios tested (Fig S3A–C). Of note, control antibodies carrying a nonbinding Fab arm and either the CD3-specific or the 5T4-specific Fab arm of DuoBody-CD3x5T4 (bsIgG1-CD3xctrl or bsIgG1-ctrlx5T4) did not induce tumor cell kill (Fig S3B), indicating that cross-linking of T cells with 5T4-expressing tumor cells is required to induce cytotoxicity.

Kinetic studies demonstrated that DuoBody-CD3x5T4 induced efficient T cell–mediated cytotoxicity of 5T4⁺ MDA-MB-231 breast cancer cells after 48 h, with maximal kill and maximum T-cell activation observed after 72-h incubation (Fig S3D and E). Tumor cell kill was associated with cytokine, granzyme B (GZMB), and perforin production (Fig S3F–H), with higher maximum levels after prolonged incubation, except for TNFα, which showed the highest

levels after 24 h (Fig S3F). T cell–mediated cytotoxicity was also associated with T-cell proliferation, as demonstrated using CFSE-labeled T cells, with the highest T-cell expansion index observed for CD8⁺ T cells (Fig S3I).

### Naive and memory CD4⁺ or CD8⁺ T cells mediate DuoBody-CD3x5T4–induced tumor cell kill

Memory T cells have been reported to respond faster and more vigorously to TCR stimulation than naive T cells, as demonstrated by more rapid secretion or production of IFNγ, IL-2 (Cho et al, 1999), and GZMB (Grossman et al, 2004). Therefore, we hypothesized that memory T cells may also respond more efficiently and faster to treatment with DuoBody-CD3x5T4 in the presence of 5T4⁺ tumor cells than naive T cells.

A T cell–mediated cytotoxicity assay was performed using CD4⁺ and CD8⁺ naive and memory T-cell subsets (Figs 1A and S4A) enriched from healthy donor human peripheral blood mononuclear cells (huPBMCs) (Fig S4B). After 24 h of incubation with DuoBody-CD3x5T4, memory CD4⁺ and CD8⁺ T-cell subsets killed MDA-MB-468 tumor cells more efficiently than their naive counterparts (Fig 1B). Nevertheless, all T-cell subsets induced complete tumor cell kill after 48 and 72 h (Fig 1B), with comparable potency after 72 h. This indicates that the different subsets are equally capable of mediating DuoBody-CD3x5T4–induced tumor cell kill, albeit with slightly different kinetics. More rapid tumor cell kill was associated with more rapid T-cell activation, as illustrated by the up-regulation of CD69 (Fig 1C and D) in both naive and memory subsets. Despite comparable capacity to induce DuoBody-CD3x5T4-dependent tumor cell kill, the frequency of activated naive T cells was lower after 24 h but higher after 48–72 h when compared with memory populations, in particular for naive CD4⁺ T cells. Also, considerable differences in cytokine production were measured in cell culture supernatants of the individual T-cell subsets. CD4⁺ memory T cells were the major producers of IFNγ, IL-6, IL-8, and TNFα, whereas the other T-cell subsets produced lower levels (Fig 1E). Strikingly, although tumor cell kill was comparable between the T-cell subsets (Fig 1B), at least twofold differences in GZMB production were observed when comparing CD4⁺ memory and CD8⁺ naive T cells to CD4⁺ naive and CD8⁺ memory T cells (Fig 1E).

Interestingly, naive CD4⁺ and CD8⁺ T cells displayed increased expression of the memory-associated marker CD45RO after 48 and 72 h of coculture with MDA-MB-468 tumor cells in the presence of DuoBody-CD3x5T4 (Fig 1F and S4C), indicating a shift of the naive T-cell subsets toward a memory-like phenotype. In memory T-cell subsets, the percentage of CD45RO⁺ T cells remained unaffected (Fig S4D). These data suggest that DuoBody-CD3x5T4–mediated cross-linking of T cells with 5T4⁺ tumor cells induces differentiation of naive T cells toward a memory-like phenotype.

### DuoBody-CD3x5T4–induced T cell–mediated cytotoxicity and T-cell activation are dependent on the presence of 5T4⁺ target cells

Next, we evaluated the relation between 5T4 expression levels and DuoBody-CD3x5T4 cytotoxic activity in a panel of 16 tumor cell lines, derived from nine different solid tumor indications. These cell lines

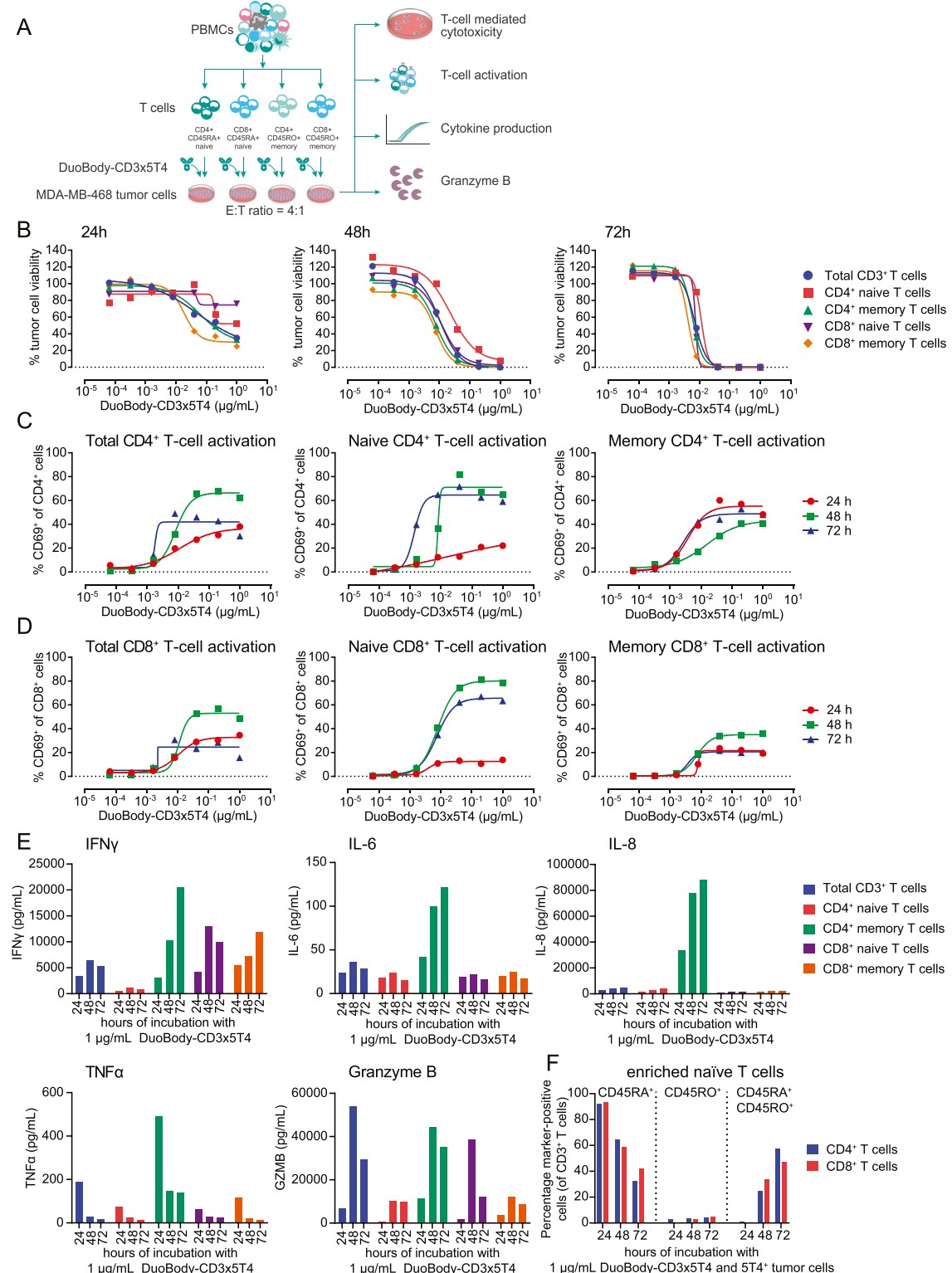

**Figure 1. Enriched naive and memory CD4⁺ and CD8⁺ T cells can mediate DuoBody-CD3x5T4–induced cytotoxicity in vitro.**
**(A)** Schematic representation of the T cell–mediated cytotoxicity assay using enriched naive and memory CD4⁺ and CD8⁺ T cells in coculture with MDA-MB-468 tumor cells. **(B)** Kinetics of DuoBody-CD3x5T4–induced cytotoxicity mediated by the indicated T-cell subsets, showing a representative donor of three donors tested. **(C, D)** Kinetics of DuoBody-CD3x5T4–induced CD4⁺ (C) and CD8⁺ (D) T-cell activation for total, naive, or memory subsets, showing a representative donor of three donors tested. **(E)** Kinetics of DuoBody-CD3x5T4–induced cytokine and GZMB production by indicated T-cell subsets, showing a representative donor of two donors tested. **(F)** The

showed a broad range of 5T4 expression with average plasma membrane 5T4 expression levels ranging from ~6,000 to ~61,500 molecules/cell (Fig 2A and Table S2). DuoBody-CD3x5T4 induced dose-dependent T cell–mediated killing of all cell lines (Figs 2B–D and S5A–L). Low levels of 5T4 expression (e.g., ~6,000 molecules/cell for RL95-2 cells) were sufficient for complete tumor cell kill (Fig 2B), although there was a statistically significant inverse correlation between half-maximal inhibitory concentrations ($IC_{50}$) and 5T4 expression (Fig 2E). Knocking out 5T4 by CRISPR-Cas9 in MDA-MB-231 cells (Fig S5M) completely abrogated DuoBody-CD3x5T4–induced T cell–mediated cytotoxicity (Fig 2F), confirming the need for 5T4 expression on the target cell.

### DuoBody-CD3x5T4–induced T cell–mediated cytotoxicity partially depends on Fas and IFNGR1 expression

The role of Fas expression in DuoBody-CD3x5T4–induced T cell–mediated cytotoxicity was evaluated using MDA-MB-231 cells, which showed up-regulation of Fas upon stimulation with IFNγ (Fig 3A). CRISPR-Cas9–mediated KO of Fas did not affect 5T4 expression (Fig 3A) and had no or only a small (albeit significant) effect on the frequency of activated CD4$^+$ and CD8$^+$ T cells in the presence of DuoBody-CD3x5T4 (Figs 3B and C and S6A). However, tumor cell kill by both T-cell subsets was reduced considerably (Fig 3D and E). These data demonstrate a significant contribution of the Fas-mediated pathway to DuoBody-CD3x5T4–induced cytotoxicity. Similar results were obtained when NCI-H1299 cells were used as target cells (Fig S6B–D).

As incubation with IFNγ up-regulates Fas expression on MDA-MB-231 cells (Fig 3A), we hypothesized that expression of the receptor for IFNγ could also play a role in the DuoBody-CD3x5T4–induced kill. The IFNγ receptor consists of two ligand binding IFNGR1 chains and two signal-transducing IFNGR2 chains (Bach et al, 1997). CRISPR-Cas9–mediated KO of *IFNGR1* did not reduce 5T4 or Fas expression (Fig 3F), but DuoBody-CD3x5T4–induced T cell–mediated kill was severely reduced when *IFNGR1* was knocked out (Fig 3G). T-cell activation was not affected (Figs 3H and S6E). This suggests that DuoBody-CD3x5T4–mediated cross-linking of T cells and tumor cells still results in efficient T-cell activation but that the tumor cells lacking IFNGR1 have become less sensitive to T cell–mediated kill in a Fas-independent manner.

These data indicate that DuoBody-CD3x5T4 does not only induce kill via the classical perforin-GZMB pathway but that the death receptor pathway (Fas up-regulated via IFNγ) and/or the IFNγ signaling pathway also play a direct role in DuoBody-CD3x5T4–induced tumor cell kill.

### DuoBody-CD3x5T4–induced bystander kill partially depends on IFNGR1 expression

We have shown that tumor cell kill induced by DuoBody-CD3x5T4 is dependent on 5T4 target expression (Fig 2F). Recent reports however have suggested that CD3 bsAbs may induce bystander kill of target-negative cells that reside in close proximity of target-positive cells (Ross et al, 2017; Upadhyay et al, 2021). To determine if DuoBody-CD3x5T4 can induce bystander kill in heterogeneous cocultures of 5T4$^+$ and 5T4$^-$ tumor cells, parental and 5T4 KO MDA-MB-231 tumor cells were mixed at different ratios and incubated with T cells and DuoBody-CD3x5T4. Strikingly, 5T4$^-$ tumor cells were killed in these heterogeneous cultures, almost to a similar level as 5T4$^+$ tumor cells (Fig 4A).

To study if bystander kill of 5T4$^-$ tumor cells was mediated by direct contact with activated T cells, parental (5T4$^+$) MDA-MB-231 cells were incubated with T cells and DuoBody-CD3x5T4 for 72 h, resulting in T cell–mediated kill of the tumor cells (Fig S7A) and T-cell activation (Fig S7B). Supernatants of the cocultures, either with or without the activated T cells but without parental tumor cells, were transferred to cultures of MDA-MB-231 5T4 KO cells and incubated for 72 h (Fig 4B). Supernatant containing DuoBody-CD3x5T4-activated T cells induced cytotoxicity of 5T4$^-$ tumor cells, in contrast to T cells that had been cocultured in the presence of the negative control bsIgG1-CD3xctrl (Fig 4C). Strikingly, supernatant containing T cells activated by general anti-CD3/CD28 stimulation also resulted in kill of 5T4$^-$ tumor cells (Fig 4C). This indicates that activated T cells can induce cytotoxicity of tumor cells, irrespective of the activating stimulus, probably because of factors secreted by these activated T cells. Indeed, cell-free supernatant derived from T cells and 5T4$^+$ cells cocultured in the presence of DuoBody-CD3x5T4 could also induce tumor cell kill but to a lesser extent than supernatant including T cells (Fig 4C), suggesting that (a) the observed bystander kill at least partially relies on a soluble component in the supernatant of activated T cells and (b) that interaction of activated T cells with 5T4$^-$ tumor cells enhances the observed bystander kill. As supernatant of activated T cells induced Fas expression on 5T4$^-$ tumor cells (Fig 4D), we hypothesized that this soluble component present in the supernatant was IFNγ.

To evaluate the role of Fas and IFNGR1 in DuoBody-CD3x5T4–induced bystander kill, 5T4/Fas and 5T4/IFNGR1 double-KO tumor cells were generated (Fig 4E). Loss of IFNGR1 inhibited the up-regulation of IFNγ-responsive proteins Fas and PD-L1 after incubation with IFNγ but did not affect baseline Fas expression (Fig 4E). Parental and 5T4, 5T4/Fas, or 5T4/IFNGR1 double-KO MDA-MB-231 tumor cells were mixed in different ratios and cocultured with purified T cells and DuoBody-CD3x5T4. Survival of 5T4/Fas double-KO cells was similar to that of the single 5T4 KO cells (Figs 4F and S7C), suggesting that Fas expression does not play a role in bystander kill. Interestingly, bystander kill of 5T4 KO cells was completely blocked when IFNGR1 was knocked out (Figs 4F and S7C), indicating that IFNγ was essential for bystander kill. T-cell activation was comparable between the different KO cell lines (Figs 4G and H and S7D), illustrating that the observed differences in bystander kill were not caused by differential T-cell activation. For some of the T-cell donors, IFNγ levels were higher in the 5T4 KO or 5T4/Fas double-KO cocultures with parental cells than in the 5T4/IFNGR1 double-KO cocultures, whereas other donors showed

---

transition of naive (CD45RA$^+$RO$^-$) to memory-like (CD45RA$^+$RO$^+$) CD4$^+$ and CD8$^+$ T cells induced by 1 μg/ml DuoBody-CD3x5T4 at different time points, showing a representative donor of two donors tested. This was determined by flow cytometry as a percentage of total CD3$^+$ T cells.

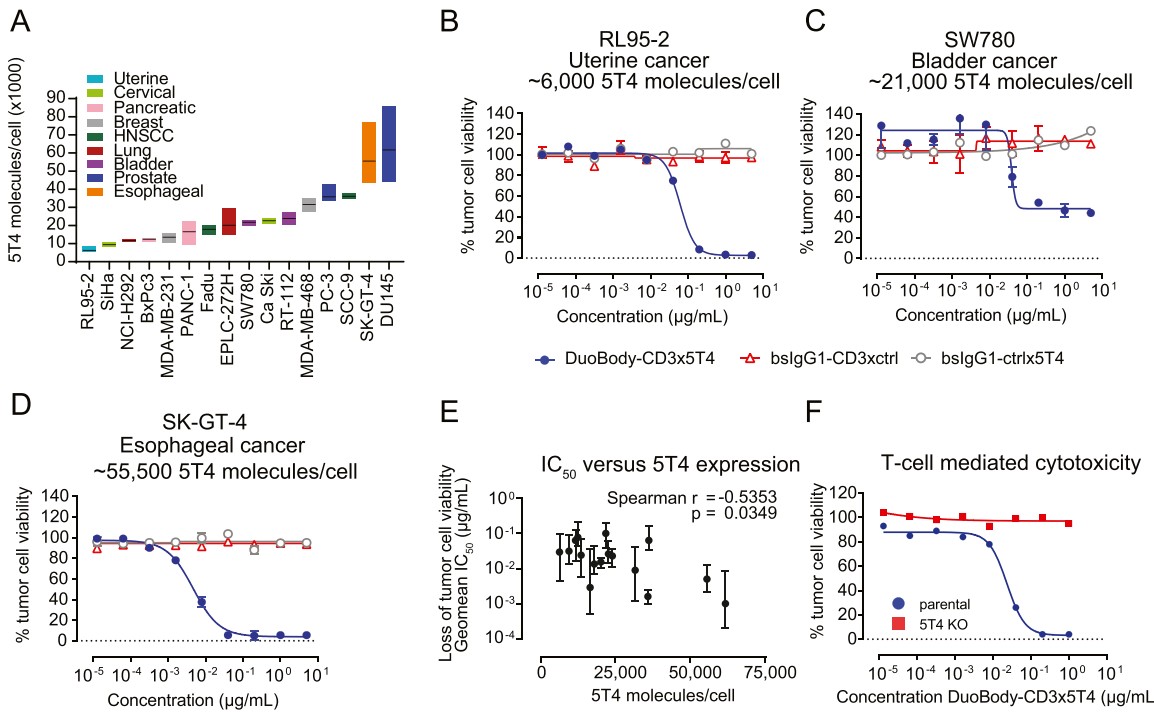

**Figure 2. DuoBody-CD3x5T4–induced T cell–mediated cytotoxicity and T-cell activation are dependent on the presence of 5T4+ target cells.**
**(A)** Expression of 5T4 in a panel of cancer cell lines of different indications, determined by quantitative flow cytometry (n = 2–4 per cell line). Shown here are the mean (horizontal line) and the range of expression detected. **(B, C, D)** Dose-dependent T cell–mediated cytotoxicity of different tumor cell lines in the presence of DuoBody-CD3x5T4 and purified T cells (E:T ratio = 4:1) after 72 h. Shown are mean survival percentages ± SD of duplicate wells derived from a representative donor of five (RL95-2) or three (SW780, SK-GT-4) donors tested. **(E)** Geomean $IC_{50}$ values (and range) for DuoBody-CD3x5T4–induced loss of tumor cell viability of all tested tumor cell lines (n = 3–10 donors/cell line) plotted against the number of 5T4 molecules/cell ($P ≤ 0.05$; nonparametric Spearman correlation). **(F)** T cell–mediated cytotoxicity of MDA-MB-231 parental and 5T4 KO cells in the presence of DuoBody-CD3x5T4 and purified T cells (E:T ratio = 4:1) after 72 h, showing a representative donor of 4 donors tested.

comparable IFNγ levels in all three different cocultures, suggesting that these differences could be donor-specific (Figs 4I and S7E).

## DuoBody-CD3x5T4 induces antitumor activity in vivo

Next, we studied the antitumor activity of DuoBody-CD3x5T4 in humanized mouse models. First, we analyzed the antitumor activity of DuoBody-CD3x5T4 by subcutaneous (SC) co-engraftment of NOD-SCID mice with MDA-MB-231 breast cancer cells and huPBMCs. In both the prophylactic (single dose directly after co-engraftment, Fig S8A–E) and the therapeutic (weekly dosing after tumor establishment, Fig S8F–J) setting, treatment with DuoBody-CD3x5T4 (at all dose levels) resulted in inhibition of tumor outgrowth in individual mice, with dose levels of 0.5 and 5 mg/kg significantly prolonging progression-free survival in the prophylactic setting (Fig S8E) compared with the control group.

Second, the antitumor activity of DuoBody-CD3x5T4 was studied in a prostate cancer cell line–derived xenograft (CDX) model (DU-145) in which huPBMCs were injected IV after SC tumor establishment (Fig 5A). Flow cytometry analysis of the (untreated) tumors revealed CD3+ T-cell infiltration 14 d after huPBMC administration, which increased even further after 21 d (Fig S8K). DuoBody-CD3x5T4 treatment at doses of 0.5, 5, or 20 mg/kg, initiated 7 d after huPBMC administration, induced complete tumor regression in 9/15, 13/15, and 13/15 animals, respectively (Fig 5B–D). Accordingly, progression-free

survival was significantly prolonged by DuoBody-CD3x5T4 treatment at all tested dose levels (Fig 5E).

Lastly, the antitumor activity of DuoBody-CD3x5T4 was studied in CD34+ hematopoietic stem cell (HSC)–humanized NSG mice (NSG-human immune system [HIS] mice) that were SC implanted with MDA-MB-231 breast cancer cells (Fig 5F). Treatment with DuoBody-CD3x5T4 inhibited tumor growth in a dose-dependent manner, resulting in significantly lower tumor volumes on day 42 (the last day where all treatment groups were complete, i.e., before the first tumor-related death; Fig 5G–J) and significantly prolonged progression-free survival when compared with IgG1-ctrl (Fig 5K). Similar results were obtained in a lung cancer patient-derived xenograft (PDX) model in NOG-HIS mice, where 5 mg/kg DuoBody-CD3x5T4 significantly extended progression-free survival (Fig S8L–P).

## DuoBody-CD3x5T4 induces markers of T-cell activation in peripheral blood and the tumor microenvironment in tumor-bearing mice

To study peripheral and intratumoral PD biomarkers in response to DuoBody-CD3x5T4, NSG-HIS mice bearing MDA-MB-231–derived tumors were treated with DuoBody-CD3x5T4. Blood samples were taken at 24, 48, and 72 h, and tumor tissue was evaluated after 72 h (Fig 6A). In peripheral blood, dose-dependent T-cell activation, measured by the up-regulation of activation markers CD69, CD25,

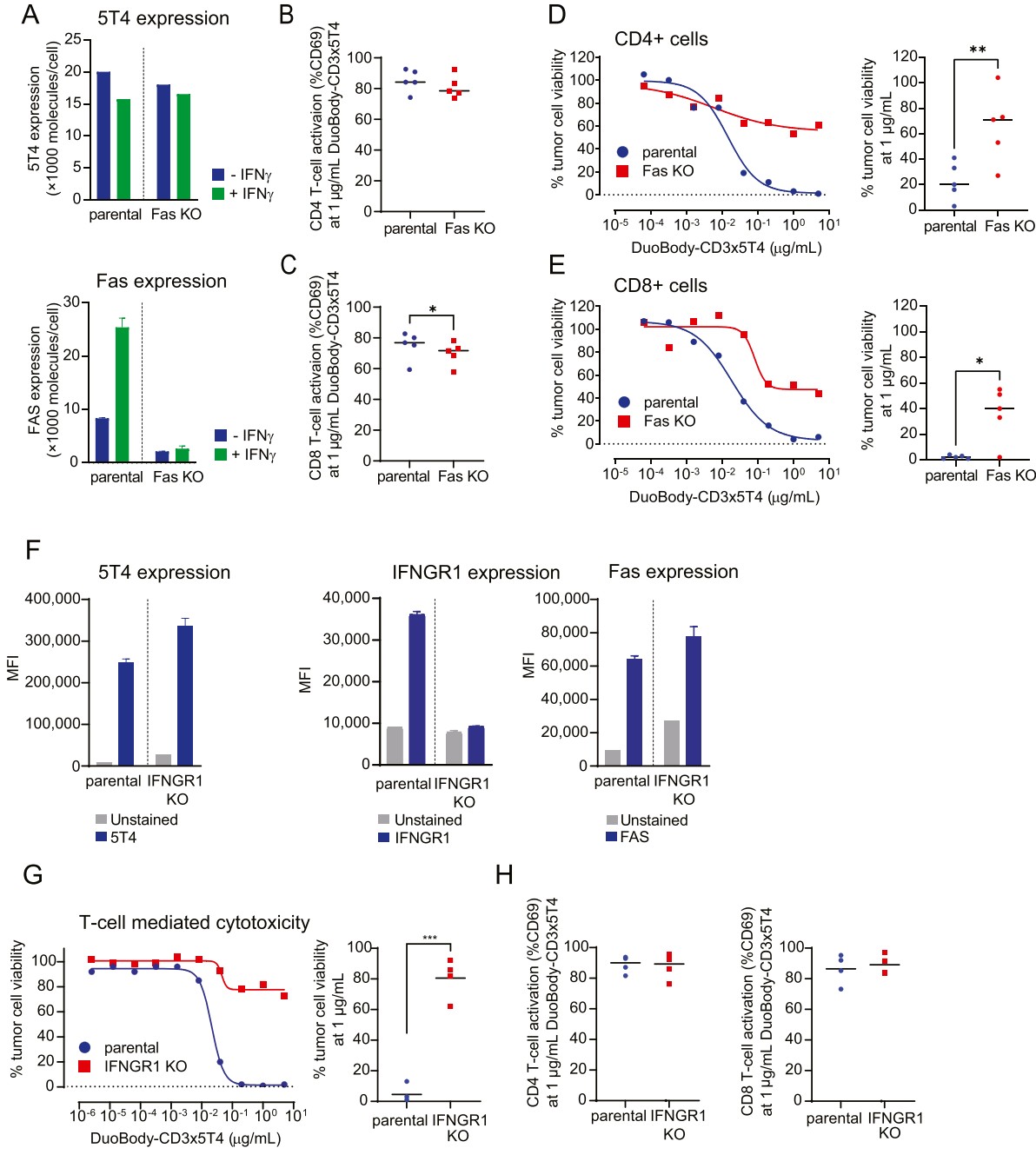

**Figure 3. Fas and IFNGR1 expression contribute to the induction of T cell–mediated cytotoxicity by DuoBody-CD3x5T4.**
**(A)** 5T4 and Fas expression on MDA-MB-231 parental and Fas KO cells was measured by quantitative flow cytometry with or without prior overnight incubation with IFNγ (100 ng/ml). **(B, C, D, E)** MDA-MB-231 parental and Fas KO cells were incubated with purified CD4+ (B, D) or CD8+ (C, E) T cells (E:T ratio = 4:1, n = 5 donors) and DuoBody-CD3x5T4 for 72 h. T-cell activation (B, C) and T cell–mediated cytotoxicity (D, E) were analyzed. **(B, C)** Activation of T cells from five donors at 1 μg/ml DuoBody-CD3x5T4 (*$P \leq$ 0.05, paired $t$ test). **(D, E)** The left panels show dose-dependent T cell–mediated cytotoxicity in a representative experiment, and the right panels show tumor cell viability at 1 μg/ml DuoBody-CD3x5T4 from five donors (*$P \leq$ 0.05 and **$P \leq$ 0.01, paired $t$ test). **(F)** 5T4, IFNGR1 and Fas expression on MDA-MB-231 parental and IFNGR1 KO cells was measured by flow cytometry. **(G, H)** MDA-MB-231 parental and IFNGR1 KO cells were incubated with purified T cells (E:T ratio = 4:1, n = 4 donors) and DuoBody-CD3x5T4 for 72 h. **(G)** The left panel shows dose-dependent T cell–mediated cytotoxicity in a representative experiment, and the right panel shows tumor cell viability at 1 μg/ml DuoBody-CD3x5T4 from four donors (***$P \leq$ 0.005, paired $t$ test). **(H)** T-cell activation at 1 μg/ml DuoBody-CD3x5T4 from four donors.

and PD-1, was observed in CD8+ (and to a lesser extent and only for CD69 in CD4+) T cells at 72 h after treatment with DuoBody-CD3x5T4 (Figs 6B and S9A). Although activated intratumoral T cells were observed in both treated and control animals, relative expression

of PD-1 on intratumoral CD8+ T cells and CD25 on CD4+ T cells was increased after treatment with DuoBody-CD3x5T4 in a dose-dependent manner (Figs 6C and S9B). Peripheral blood IFNγ, IL-6, and IL-8 levels also showed a dose-dependent increase, which was

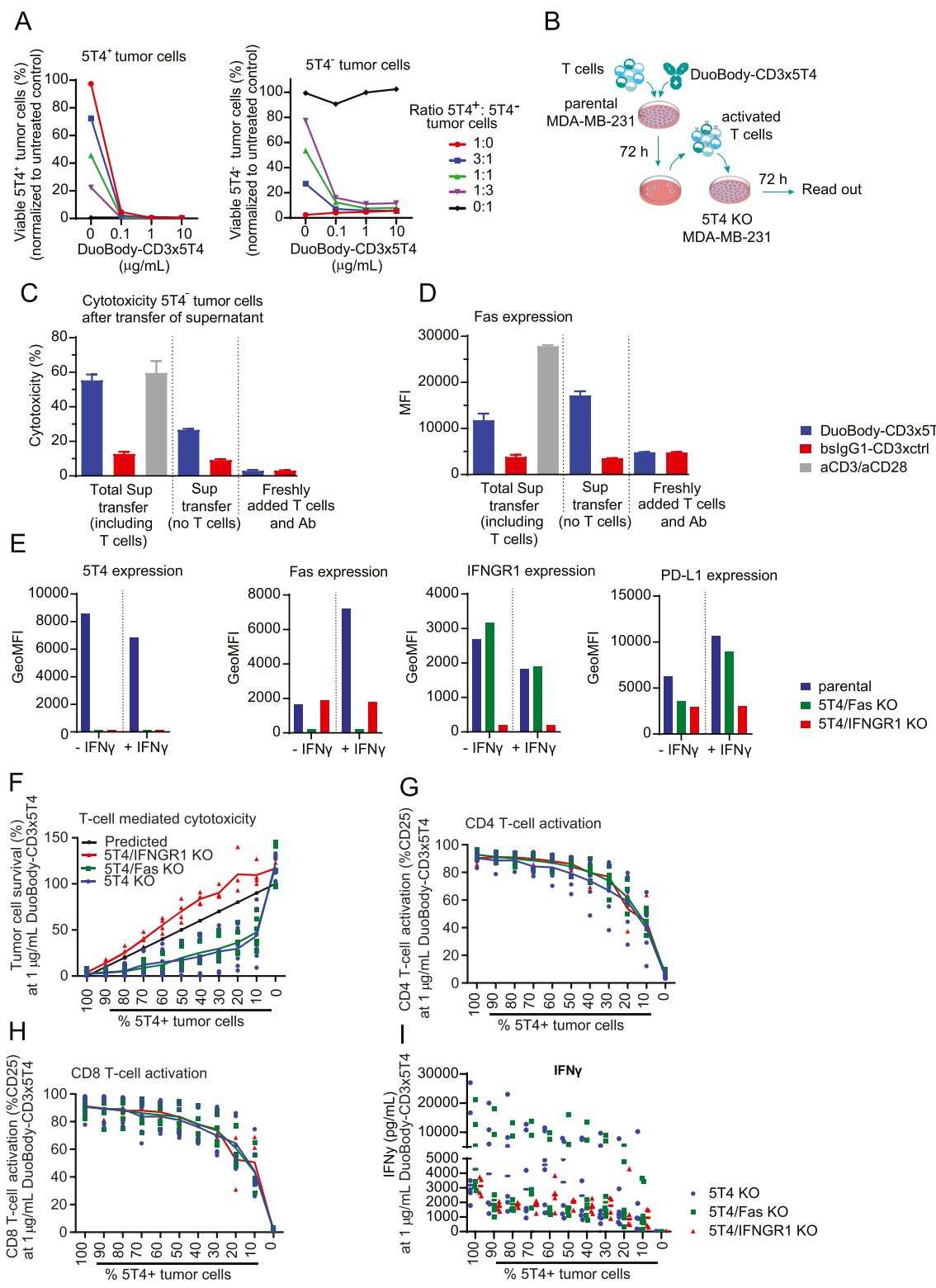

**Figure 4. DuoBody-CD3x5T4 can induce bystander kill which is dependent on IFNGR1 expression.**
**(A)** Parental and 5T4 KO MDA-MB-231 tumor cells were mixed in different ratios, as indicated, and incubated with purified T cells (E:T ratio = 4:1, n = 2 donors) and DuoBody-CD3x5T4 for 72 h. T cell–mediated cytotoxicity of 5T4⁺ (left panel) and 5T4⁻ (right panel) tumor cells was determined by flow cytometry, showing a representative donor of three donors tested. **(B, C, D)** Parental (5T4⁺) MDA-MB-231 tumor cells were cocultured with purified T cells (E:T = 4:1, n = 2 donors) and incubated with 10 μg/ml DuoBody-CD3x5T4 or bsIgG1-CD3xctrl for 72 h. As a positive control, T cells were incubated with anti-CD3/CD28 beads (but without tumor cells) for 72 h. **(B)** The supernatant (either with or without T cells) was transferred to MDA-MB-231 5T4 KO cells and incubated for 72 h (B). As negative control, MDA-MB-231 5T4 KO cells were

particularly evident at 24 h after DuoBody-CD3x5T4 treatment (Fig 6D). Cytokine levels were still elevated at 48 h after treatment but had returned to baseline after 72 h. TNFα levels were below the detection limit in most of the samples and are therefore not shown. GZMB was not elevated in the blood after DuoBody-CD3x5T4 treatment.

To monitor intratumoral changes associated with antitumor response, T-cell infiltration and T-cell activation in tumor tissue were quantified in the same mouse model (NSG-HIS mice with established MDA-MB-231 tumors) at 72 h after treatment. DuoBody-CD3x5T4 treatment induced plasma cytokine production and intratumoral T-cell activation as described above (Fig S9C and D). Although individual mice showed an increase in the frequency of CD3⁺ T cells and CD25⁺ cells 72 h after DuoBody-CD3x5T4 treatment, there was no statistically significant difference in the frequency of CD3⁺ or CD25⁺ cells (Fig 6E and F) or in the spatial distribution of CD3⁺ cells within the tumors (Fig S9E) between treatment and control groups. This may be explained by the early time point after treatment and by the presence of predominantly CD4⁺ T cells in the tumors (Fig S9F), whereas only intratumoral CD8⁺ T cells showed CD25 up-regulation as measured by flow cytometry (Fig S9D). Importantly, DuoBody-CD3x5T4 did induce the frequency of Ki67⁺ CD3⁺ T cells (Fig 6E and F), indicative of intratumoral T-cell proliferation in response to treatment. GZMB staining was also significantly increased in DuoBody-CD3x5T4–treated tumors (Fig 6E and F), suggesting induction of intratumoral T cell–mediated cytotoxicity. Quantification of GZMB in tumor lysates confirmed increased production in the tumor microenvironment after DuoBody-CD3x5T4 treatment (Fig 6G). Moreover, intratumoral IFNγ levels were significantly increased in DuoBody-CD3x5T4 treated animals, whereas intratumoral IL-6, IL-8, and TNFα levels showed a trend of increase that was not statistically significant when compared with IgG1-ctrl-treated tumors (Fig 6G).

In summary, antitumor activity in response to DuoBody-CD3x5T4 treatment was associated with peripheral blood PD markers, including T-cell activation and increased plasma cytokine levels, in addition to intratumoral PD markers such as T-cell activation, T-cell proliferation, and production of IFNγ and GZMB in the tumor.

### DuoBody-CD3x5T4 induces T cell–mediated tumor cell killing in patient-derived solid tumor samples ex vivo

Finally, we explored the capacity of DuoBody-CD3x5T4 to induce T cell–mediated tumor cell kill and T-cell activation in an autologous setting, using single-cell suspensions from four solid tumor biopsies that contained tumor cells and tumor-infiltrating lymphocytes (TILs, Fig 7A). For two of the biopsies, tissue sections were available for immunohistochemistry (IHC) analysis, and expression of 5T4 and the presence

of CD3⁺ T cells were confirmed (Fig 7B). Flow cytometry analysis of the single-cell suspensions demonstrated 5T4 expression on tumor cells of three (OV1, UT1, and OV2) of four biopsies (Fig 7C). The CD3⁺ T-cell percentage ranged from 9 to 45% (Fig 7D). DuoBody-CD3x5T4 induced dose-dependent, autologous TIL-mediated tumor cell killing in all three 5T4⁺ tumor samples ex vivo, whereas the 5T4⁻ tumor sample (LU1) was not sensitive to treatment (Fig 7E). DuoBody-CD3x5T4–induced TIL-mediated tumor cell kill coincided with increased TIL activation, as shown by induction of the T-cell activation marker CD25 (Fig 7F). DuoBody-CD3x5T4 induced no or minimal changes in PD-1 expression on the TILs, which may be related to high baseline PD-1 expression (~40–70%; Fig 7F), indicative of an exhausted T-cell phenotype, in three of four samples. DuoBody-CD3x5T4–induced TIL-mediated tumor cell kill was also associated with dose-dependent production of IFNγ (Fig 7G) and GZMB (Fig 7H); no IFNγ and GZMB production was observed when DuoBody-CD3x5T4 was incubated with the 5T4⁻ LU1 tumor sample, in line with the absence of kill.

In summary, DuoBody-CD3x5T4 could engage autologous TILs to mediate killing of 5T4⁺ tumor cells in patient-derived solid tumor samples ex vivo, even when a large percentage of TILs expressed PD-1 at baseline.

## Discussion

Here, we describe mechanistic and pharmacodynamic studies performed to obtain an in-depth preclinical understanding of CD3 bsAbs in solid cancers, using DuoBody-CD3x5T4, a CD3 bsAb that cross-links T cells with 5T4-expressing tumor cells to induce tumor cell kill. DuoBody-CD3x5T4 induced T cell–mediated cytotoxicity in vitro and antitumor activity in humanized CDX and PDX mouse models in vivo. Treatment with DuoBody-CD3x5T4 was accompanied by T-cell activation, T-cell proliferation, and production of inflammatory cytokines, GZMB, and perforin. Naive and memory T cells, both CD4⁺ and CD8⁺, showed equal capacity to induce tumor cell kill in vitro, although it should be noted that memory T cells were evaluated without further distinction between effector and central memory subsets. This finding is intriguing as memory CD8⁺ T cells have been generally assumed to be the main effectors of T cell–mediated cytotoxicity. Recent publications however have shown that CD4⁺ T cells can also function as cytotoxic T cells, both in vitro (Grossman et al, 2004; Porakishvili et al, 2004; Engelberts et al, 2020) and in vivo (Benonisson et al, 2019). Interestingly, GZMB secretion was more pronounced for CD4⁺ than for CD8⁺ memory T cells. This is in line with a previous study reporting slightly higher levels of extracellular GZMB secreted by activated memory CD4⁺ T cells compared with memory CD8⁺ T cells (Lin et al, 2014). In our in vivo studies, peripheral and intratumoral T-cell activation 72 h postdose was detected mainly in CD8⁺ T cells. This suggests that

---

incubated with fresh T cells and indicated antibodies for 72 h. **(C, D)** T cell–mediated cytotoxicity (C) and Fas expression (D) of MDA-MB-231 5T4 KO cells were determined by flow cytometry. A representative donor of two donors tested is shown. **(E)** 5T4, Fas, IFNGR1, and PD-L1 expression on MDA-MB-231 parental, 5T4/Fas KO, and 5T4/IFNGR1 KO cells was measured by flow cytometry with or without prior overnight incubation with IFNγ (100 ng/ml). **(F, G, H, I)** Parental and 5T4/Fas KO or 5T4/IFNGR1 MDA-MB-231 tumor cells were mixed in different ratios, as indicated, and incubated with purified T cells (E:T ratio = 4:1, n = 4–6) and 1 μg/ml DuoBody-CD3x5T4 for 72 h, after which T cell–mediated cytotoxicity (F), CD4⁺ (G) and CD8⁺ (H) T-cell activation, and IFNγ production (I) were analyzed. The predicted black line in (F) refers to the outcome of the assay when no bystander killing is expected, for example, if 50% of tumor cells are 5T4⁺, only 50% of tumor cells will be killed.

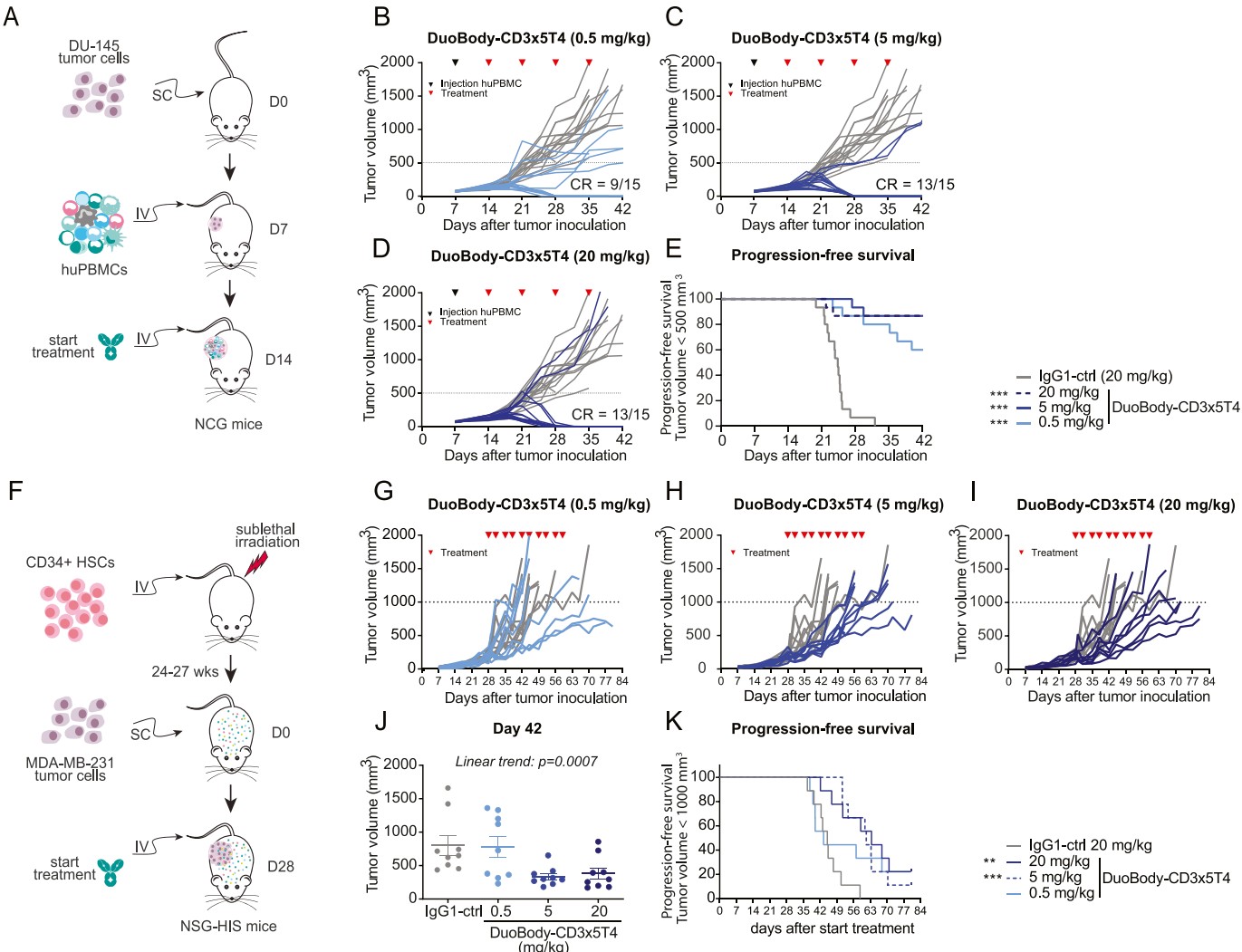

**Figure 5.  DuoBody-CD3x5T4 demonstrates antitumor activity in vivo.**
**(A, B, C, D, E)** When SC-implanted prostate cancer CDX (DU-145) tumors reached a volume of ~75 mm³, huPBMCs were injected IV. **(A)** 7 d later, treatment with DuoBody-CD3x5T4 (0.5, 5, or 20 mg/kg; IV) or IgG1-ctrl (20 mg/kg) was initiated (A; n = 15 per treatment group). **(B, C, D)** Tumor volume for individual mice in the 0.5 (B), 5 (C), or 20 (D) mg/kg DuoBody-CD3x5T4 treatment groups over time. The gray lines indicate tumor volumes of individual mice receiving 20 mg/kg IgG1-ctrl. The dotted line indicates the cutoff for progression-free survival (500 mm³). **(E)** Progression-free survival, defined as the percentage of mice with tumor volume <500 mm³, is shown as a Kaplan–Meier curve. Mantel–Cox analysis with Bonferroni correction for multiple comparisons was used to compare progression-free survival between treatment groups and control, with ***$P \leq 0.001$. **(F, G, H, I, J, K)** The breast cancer MDA-MB-231 CDX model was established by SC implantation of $5 \times 10^6$ MDA-MB-231 cells into NSG-HIS mice. **(F)** When tumors reached an average volume of ~200 mm³, mice were treated with DuoBody-CD3x5T4 (0.5, 5, and 20 mg/kg, n = 9/group) or IgG1-ctrl (20 mg/kg, n = 9) (F). **(G, H, I)** Tumor volume for individual mice in the 0.5 (G), 5 (H), or 20 (I) mg/kg DuoBody-CD3x5T4 treatment groups over time. The gray lines indicate tumor volumes of individual mice receiving 20 mg/kg IgG1-ctrl. The dotted line indicates the cutoff for progression-free survival (500 mm³). **(J)** Tumor volumes of the different groups on the last day (day 42 after tumor inoculation) where all groups were complete. An ordinary one-way ANOVA with a posttest for linear trend was used to compare log-transformed tumor volumes ($P = 0.0007$). **(K)** Progression-free survival, defined as the percentage of mice with tumor volume <1,000 mm³, is shown as a Kaplan–Meier curve. Mantel–Cox analysis with Bonferroni correction for multiple comparisons was used to compare progression-free survival between treatment groups and control, with **$P \leq 0.01$ and ***$P \leq 0.001$.

although both CD4⁺ and CD8⁺ T cells can mediate CD3 bsAb-mediated tumor cell kill, the dynamics of CD4⁺ and CD8⁺ T-cell activation or the distribution of CD4⁺ and CD8⁺ T cells may differ between in vitro and in vivo settings.

In cocultures with 5T4⁺ tumor cells, DuoBody-CD3x5T4 induced at least partial differentiation of naive T cells into a memory-like phenotype in vitro, hinting to potential maturation in response to treatment. We hypothesized that these CD45RA⁺/CD45RO⁺ T cells are in the middle of their transition from a naive phenotype

(CD45RA⁺) to a more memory-like phenotype (CD45RO⁺). Longer incubation might have resulted in a full transition into a single CD45RO⁺ population. The finding that naive T cells transition into a memory-like phenotype is in line with a phase I study with REGN5458, a BCMA-targeting CD3 bsAb, where an increase of peripheral effector memory T cells was found (Cooper et al, 2019). An increased frequency of memory(-like) T cells in response to DuoBody-CD3x5T4 treatment might therefore serve as a PD marker. Of note, the observed in vitro T-cell differentiation is more likely to

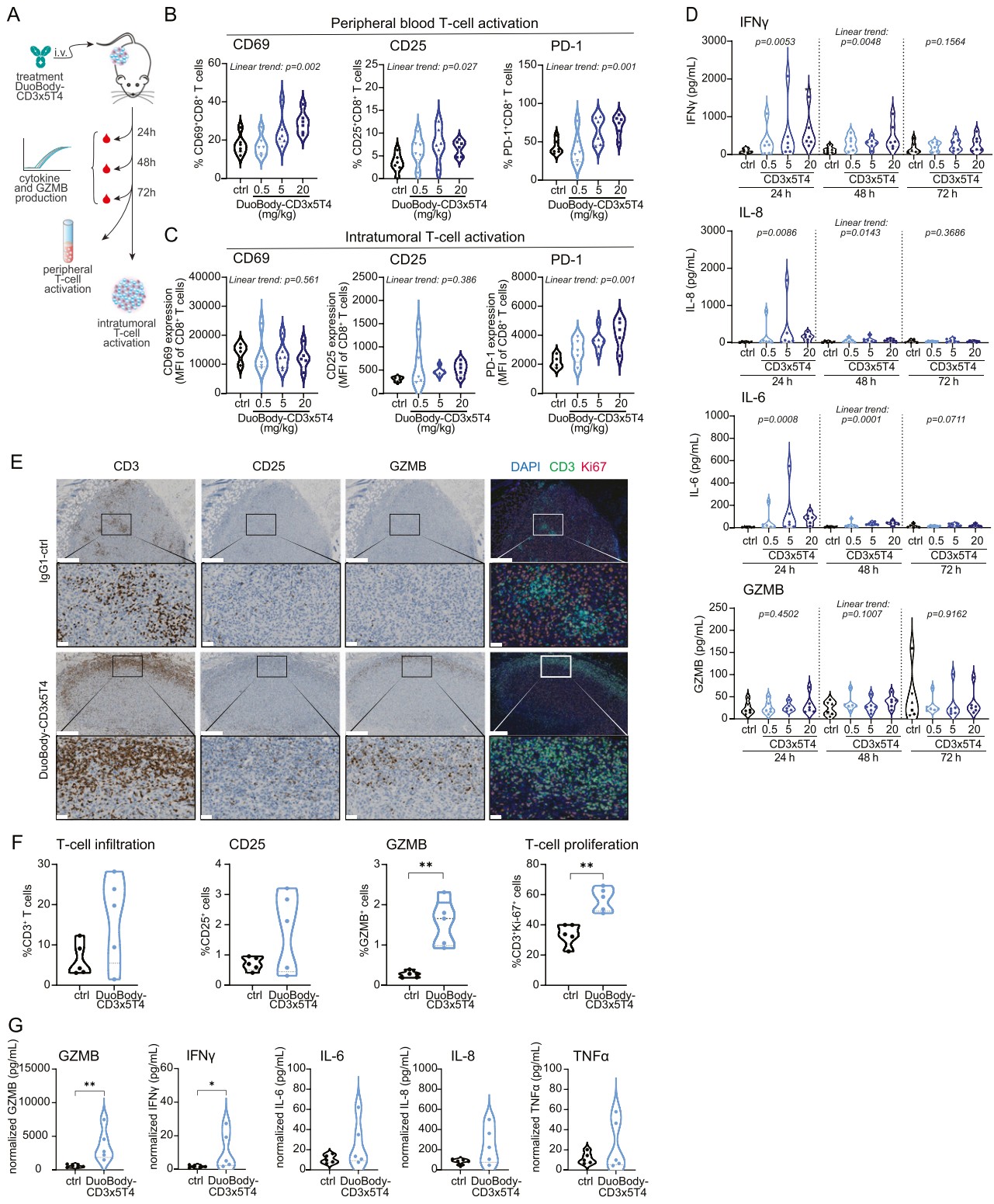

**Figure 6. Antitumor efficacy of DuoBody-CD3x5T4 is associated with peripheral and intratumoral T-cell activation.**
**(A, B, C, D)** The breast cancer MDA-MB-231 CDX model was established by SC implantation of 5 × 10$^6$ MDA-MB-231 cells into NSG-HIS mice (as described in Fig 5F). **(A)** When tumors reached an average volume of ~200 mm$^3$, mice were treated with DuoBody-CD3x5T4 (0.5, 5, and 20 mg/kg, n = 9/group) or IgG1-ctrl (20 mg/kg, n = 9). Blood samples were taken at 24, 48, and 72 h after treatment. After 72 h, tumors were excised, dissociated, and analyzed for T-cell activation by flow cytometry. **(B, C)** CD8$^+$ T-cell activation in the blood (B) and the tumor (C) was determined by flow cytometry. **(B)** Shown is the percentage of CD69$^+$, CD25$^+$, or PD-1$^+$ CD8$^+$ T cells. Ordinary one-way ANOVA with posttest for linear trend was used to compare the percentage activation marker–positive cells between the different treatment groups (CD69$^+$, P = 0.002; CD25$^+$, P = 0.027; PD-1$^+$, P = 0.001). **(C)** Shown are the relative expression levels (mean fluorescence intensity) of CD69, CD25, and PD-1 on intratumoral CD8$^+$ T cells. Ordinary one-way ANOVA

be a consequence of artificial CD3 cross-linking by DuoBody-CD3x5T4 in the presence of 5T4+ tumor cells, rather than actual antigen-specific memory formation. This is in line with preclinical studies using CD3xTA99, a mouse-specific gp75-targeting CD3 bsAb, which was unable to induce immunological memory in a syngeneic mouse model (Benonisson et al, 2019). Nevertheless, DuoBody-CD3x5T4-mediated maturation of T cells could have a pro-inflammatory effect on the tumor microenvironment and might increase antitumor activity, even if T-cell maturation would occur in a non–antigen-specific manner. In-depth clinical biomarkers analyses are needed to understand this process.

The Fas pathway was shown to significantly contribute to DuoBody-CD3x5T4–induced T cell–mediated kill of 5T4+ tumor cells, in line with recent data implicating Fas and its ligand in antigen-specific T cell–mediated cytotoxicity, including T cell–mediated kill induced by the CD3xCD19 bsAb blinatumomab (Upadhyay et al, 2021). We and others (Upadhyay et al, 2021) have shown that IFNγ exposure can up-regulate Fas expression on tumor cells. Because IFNγ, produced in response to T-cell activation, can modulate distant tumor cells (Hoekstra et al, 2020; Thibaut et al, 2020)—including tumor cells more than 800 $\mu m$ away (Hoekstra et al, 2020)—IFNγ is likely to play a crucial role in Fas-mediated kill of tumor cells. Indeed, loss of the receptor for IFNγ (IFNGR1) in tumor cells abrogated DuoBody-CD3x5T4–induced tumor cell kill. This is in line with previous reports showing that downmodulation of JAK1 and JAK2, both essential tyrosine kinases in the IFNγ-signaling pathway, in tumor cells results in resistance to CD3 bsAbs (Arenas et al, 2021; Liu et al, 2021). The role of the IFNγ-signaling pathway in tumor cell kill by CD3 bsAb targeting is most likely not restricted to the regulation of Fas expression because IFNγ signaling is also involved in various other facets of T cell–mediated cytotoxicity, for example, in regulation of the expression of immune checkpoints (e.g., PD-L1) and the capacity of cells to present antigens by MHC complex class I (e.g., B2M). Interestingly, although low levels of 5T4 expression are sufficient for complete tumor cell kill of certain tumor cell lines (e.g., the RL95-2 uterine cancer cell line; Fig 2B), DuoBody-CD3x5T4 treatment of certain other tumor cell lines, including some cell lines with higher 5T4 expression levels (e.g., the SW780 bladder cancer cell line; Fig 2C), resulted in only partial tumor cell kill. The reason for this partial resistance to DuoBody-CD3x5T4–induced tumor cell kill is not fully understood but might involve differential sensitivity of the tumor cells to Fas or IFNγ signaling. This could potentially also play a role in the partial kill observed in the 5T4+-dissociated patient-derived solid tumor samples ex vivo, although the latter could also be explained by differences in accessibility between 2D and 3D cultures and a lower E:T ratio in the ex vivo experiments (Fig 7E).

We demonstrated that DuoBody-CD3x5T4 can induce bystander killing of 5T4− tumor cells, which (at least in part) depends on the presence of activated T cells and soluble factors produced by these cells, including IFNγ. Indeed, IFNγ signaling was shown to play an important role in bystander killing induced by DuoBody-CD3x5T4, as shown for other CD3 bsAbs (Ross et al, 2017). Surprisingly, knocking out Fas did not affect bystander killing in this in vitro system, which is contradictive to a previous report showing that Fas-FasL interaction is essential for bystander killing of antigen-negative cells by CD3 bsAbs and CAR T cells (Upadhyay et al, 2021). Intriguingly, in this report, it was shown that tumoral Fas expression at baseline was a better predictor of response in a CD19 CAR-T trial than the targeted tumor-associated antigen, CD19, itself, which was suggested to be attributed to the role of Fas in bystander killing (Upadhyay et al, 2021). As Fas expression did not influence the bystander kill in our in vitro model, direct effects of IFNγ exposure are probably implicated in the observed bystander kill. IFNγ can inhibit proliferation of tumor cells (Bromberg et al, 1996; Matsushita et al, 2015) and even trigger apoptosis by up-regulating molecules that promote apoptosis, for example, caspase-1 (Chin et al, 1997; Detjen et al, 2001). The physiological relevance of DuoBody-CD3x5T4–induced bystander killing; its potential impact on safety and efficacy; and the role of IFNγ, IFNGR1, and Fas in this process will need to be explored in an in vivo setting, where local concentrations of IFNγ are expected to be lower because of diffusion.

Importantly, DuoBody-CD3x5T4 redirected autologous TILs to mediate tumor cell kill in dissociated patient-derived solid tumor samples ex vivo, even when a large percentage of TILs expressed the exhaustion marker PD-1. Interestingly, the dissociated tumor sample in which TILs showed the lowest baseline level of PD-1 expression (OV1) displayed the most efficient kill (Fig 7E and F), suggesting that the exhaustion status of TILs could play a role in the biological activity of DuoBody-CD3x5T4. At this time, DuoBody-CD3x5T4 is not undergoing clinical development because of challenges in optimizing the benefit/risk profile.

In summary, we have demonstrated via in-depth mechanistic studies that DuoBody-CD3x5T4 efficiently induces T cell–mediated cytotoxicity of 5T4+ tumor cells and bystander kill of 5T4− tumor cells. DuoBody-CD3x5T4 antitumor activity was associated with peripheral and intratumoral T-cell activation. Lastly, even with most of the T cells expressing PD-1, DuoBody-CD3x5T4 could still induce tumor cell kill.

# Materials and Methods

### In silico analysis of 5T4 mRNA expression in different tumor indications

TPBG/5T4 mRNA expression in different tumor indications was analyzed in silico with Array Viewer (Omicsoft, V10.0) software, using

---

with posttest for a linear trend was used to compare the mean fluorescence intensity between the different treatment groups (CD69, $P = 0.561$; CD25, $P = 0.386$; PD-1, $P = 0.001$). **(D)** Peripheral blood cytokine (IFNγ, IL-6, and IL-8) and GZMB levels were measured with a multiplex MSD assay. For the ordinary one-way ANOVA with posttest for a linear trend, cytokine concentrations were log-transformed. A significant trend was observed for IFNγ, IL-6, and IL-8 at 24 and 48 h ($P ≤ 0.05$). **(E, F, G)** Similar experiment as performed in (A); mice were treated with 0.5 mg/kg DuoBody-CD3x5T4 or IgG1-ctrl (n = 5/group) after a tumor volume of 300 mm³ was reached. Blood samples were taken 24 and 48 h after treatment (Fig S9C). **(E, F)** Tumors were analyzed 72 h after treatment by IHC and IF for T-cell infiltration (CD3+), activation (CD25+ and GZMB+), and proliferation (CD3+Ki67+). Scale bars in (E) correspond to 500 $\mu m$ (upper rows) and 50 $\mu m$ (lower rows). **(G)** Tumors were dissociated 72 h after treatment and the supernatant of the dissociated tumor cells was analyzed for the presence of GZMB and cytokines by a Luminex assay. Groups were compared using Mann–Whitney, with *$P ≤ 0.05$ and **$P ≤ 0.01$.

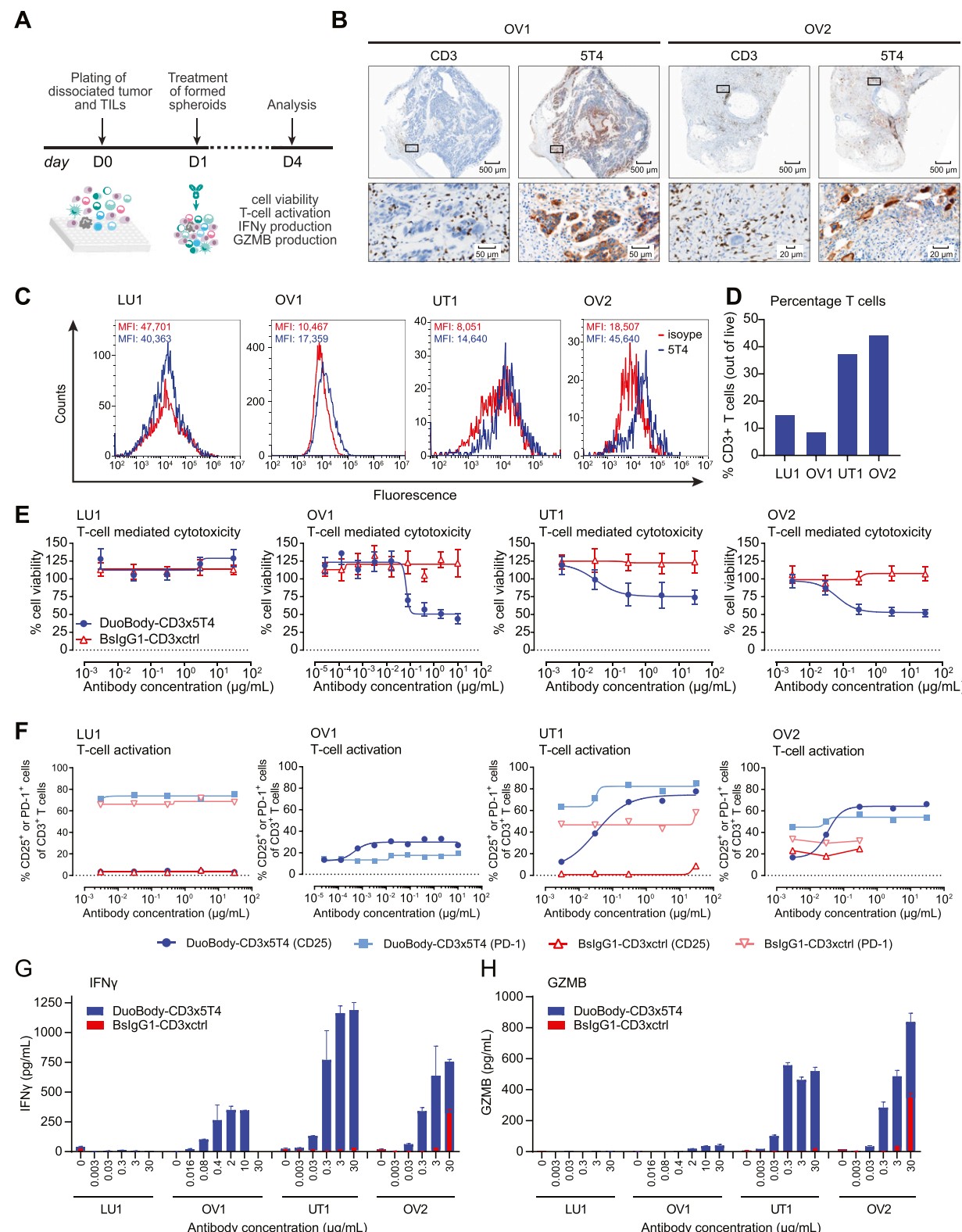

**Figure 7. DuoBody-CD3x5T4 induces T cell–mediated cytotoxicity in dissociated patient-derived solid tumor samples ex vivo.**
**(A)** Experimental outline of the evaluation of T cell–mediated tumor cell killing by DuoBody-CD3x5T4 in dissociated patient-derived solid tumor samples ex vivo. **(B)** 5T4 and CD3 expression in ovarian tumor samples OV1 and OV2, as determined by IHC. The scale bars in the top row correspond to 500 $\mu$m, and the scale bars in the bottom row correspond to 50 $\mu$m (OV1) or 20 $\mu$m (OV2). **(C)** Binding of DuoBody-CD3x5T4 or isotype control (bsIgG1-CD3xctrl) to dissociated patient-derived solid tumor cells (CD45$^-$ population) was evaluated by flow cytometry. LU, lung cancer; OV, ovarian cancer; UT, uterine cancer. **(D)** The percentage of CD3$^+$ T cells in the dissociated patient-

the The Cancer Genome Atlas mRNA expression database (National Institute of Health).

## 5T4 immunohistochemistry assay and quantification

Commercially available formalin-fixed, paraffin-embedded (FFPE) tumor tissue microarrays (TMAs; BioMax) and tumor samples OV1 and OV2 (KIYATEC) were stained with rabbit antihuman 5T4 antibody (2 µg/ml; clone EPR5529, 134162; Abcam) using the OptiView DAB IHC Detection Kit (760-700; Roche) on the Ventana BenchMark ULTRA IHC/ISH autostainer platform (Ventana Medical Systems Inc.), essentially according to manufacturer's instructions. Tumor TMA slides were scanned with standardized scanning profiles on an Axio Scan Z1 (Zeiss). Using scanned cytokeratin (CK) staining on sequential tumor TMA sections, a digital CK mask was generated for each TMA core to discriminate tumor tissue (cytokeratin-positive) from stromal tissue (cytokeratin-negative) in the scanned 5T4-stained consecutive tumor TMA section. 5T4 expression in tumor cells was digitally quantified by OracleBio (UK) using a tailored image analysis algorithm developed with HALO image analysis software (Indica Labs). 5T4 expression was scored for staining intensity (negative, low, medium, high) and percentage 5T4$^+$ cells (range 0–100%). CD3 staining on the OV1 and OV2 tumor samples was performed with 2GV6 clone at optimal prediluted (~0.4 µg/ml) concentration on the Ventana Discovery autostainer platform, using same detection as described for CD3 staining on CDX samples (see the "IHC and IF analysis on breast cancer CDX tissue sections from NSG-HIS mice" section).

## Generation of DuoBody-CD3x5T4

The CD3-targeting parental antibody was generated by germline humanization of the mouse antihuman CD3ε antibody SP34 (Pessano et al, 1985) using complementarity determining region–grafting technology and retention/back-mutation of key murine framework residues. The selected humanized variant was designated IgG1-huCACAO. For DuoBody-CD3x5T4, a variant was selected containing a histidine (H) to glycine (G) substitution at position 101 (H101G) that showed reduced affinity for human CD3ε. The fully human 5T4-targeting parental antibody IgG1-5T4-FEAR was generated by hybridoma technology after immunization of HuMAb mice with the recombinant extracellular domain (ECD) of human 5T4 (UniProt accession no. Q13641 [aa32-355]), produced by human embryonic kidney 293F (HEK293F) cells (R790-07; Invitrogen). The variable domains of both parental antibodies were sequenced and cloned in a human IgG1κ backbone containing the L234F/L235E/D265A (FEA; Eu numbering [Kabat, 1991]) Fc-silencing mutations and the K409R (5T4 parental antibody) or F405L (CD3 parental antibody) DuoBody mutations (Labrijn et al, 2013) by GeneArt Gene Synthesis (Thermo Fisher Scientific). The CD3- and 5T4-targeting parental antibodies were produced in HEK293F or Expi293F cells (A14528; Thermo Fisher Scientific), respectively, according to manufacturer's instructions, and purified from the culture supernatant by protein A

affinity chromatography. Bispecific antibodies were generated by cFAE (Labrijn et al, 2013, 2014) of the CD3- and 5T4-specific parental monoclonal antibodies. The gp120-specific antibody IgG1-b12 (Barbas et al, 1993) was used to generate nonbinding control antibodies, referred to as ctrl in this manuscript.

## Purification of PBMCs, T cells, and T-cell subsets from buffy coats

Human peripheral blood mononuclear cells (huPBMCs) were purified from human buffy coats obtained from healthy volunteers (Sanquin) by density centrifugation over a Lymphocyte Separation Medium gradient (17-829E; Lonza), according to the manufacturer's instructions. Total T cells were purified from buffy coats by negative selection, using the Rosette Separation kits (15021; STEMCELL Technologies [total T cells]), whereas CD4$^+$- and CD8$^+$-naive and memory T cells were purified from huPBMCs using EasySep Human Isolation kits (17555; STEMCELL Technologies [CD4$^+$ naive], 19157 [CD4$^+$ memory], 19258 [CD8$^+$ naive] and 19159 [CD8$^+$ memory]), all according to manufacturer's instructions.

## Cell lines and culture

The origin of the cell lines and culture media used for in vitro studies are described in Tables S3 and S4.

## Measuring binding affinity for human 5T4 and CD3ε using biolayer interferometry (BLI)

For 5T4 binding, antihuman IgG Fc capture (AHC) biosensors (18-5060; FortéBio) were preconditioned (or regenerated) by exposure to 10 mM glycine buffer, pH 1.7, for 5 s, followed by neutralization in Sample Diluent (18-1048; FortéBio) for 5 s; both steps were repeated five times. Next, AHC sensors were loaded with DuoBody-CD3x5T4 (1 µg/ml in Sample Diluent) for 600 s. After a baseline measurement in Sample Diluent (300 s), the association (200 s) and dissociation (1,000 s) of recombinant human 5T4 ECD fused to a His-tag (5T4-ECDHis; aa 32–355 of human 5T4 [UniProt accession number: Q13641] with a C-terminal His-tag, produced in house) was determined using a concentration range of 0.06–3.58 µg/ml (1.56–100 nM) using twofold dilution steps in Sample Diluent.

For CD3ε binding, Anti-Penta-HIS (HIS1K) biosensors (18-5120; FortéBio) were preconditioned (or regenerated) by exposure to 10 mM glycine buffer, pH 1.5, for 5 s, followed by neutralization in Sample Diluent for 5 s; both steps were repeated twice. The Anti-Penta-HIS biosensors were loaded with 5 µg/ml recombinant human CD3ε ECD fused to a His-tag (CD3E-ECDHis; aa 23–126 of human CD3ε [UniProt accession number: P07766] with a C-terminal His-tag, produced in house; diluted in Sample Diluent) for 1,000 s. After a baseline measurement in Sample Diluent (500 s), the association (100 s) and dissociation (1,000 s) were determined of DuoBody-CD3x5T4 (2.3–150 µg/ml [15.6–1,000 nM]), using twofold dilution steps in Sample Diluent.

---

derived solid tumor samples. **(E, F, G, H)** T cell–mediated cytotoxicity ± SEM (E), T-cell activation (F; %CD25$^+$ and PD-1$^+$), and cytokine (G) and GZMB (H) production ± SD of duplicate wells induced by DuoBody-CD3x5T4 versus control (bsIgG1-CD3xctrl) in four dissociated patient-derived solid tumor samples ex vivo.

Data were acquired using Data Acquisition Software v9.0.0.49d (FortéBio) and analyzed with FortéBio Data Analysis Software v9.0.0.12 (for 5T4) or v9.0.0.14 (for CD3ε). Data traces were corrected by subtraction of a reference curve (Sample Diluent instead of antigen [for 5T4] or antibody [for CD3ε]). The y-axis was aligned to the last 10 s of the baseline. Interstep Correction alignment to dissociation and Savitzky-Golay filtering were applied. Data traces with a response <0.05 nm were excluded from analysis. Data were fitted using the 1:1 model (Langmuir), using a global full fit with the window of interest set as the full association (200 s) and dissociation (1,000 s) times for 5T4 binding or with the association and dissociation times set at 100 s for CD3ε binding.

### Binding of DuoBody-CD3x5T4 to membrane-expressed CD3 or 5T4

To assess binding to membrane-expressed CD3, Jurkat T cells (100,000 cells/well) were resuspended in 50 μl FACS buffer (1× PBS [BE17-517Q; Lonza] supplemented with 0.1% [wt/vol] BSA [10735086001; Roche] and 0.02% [wt/vol] NaN₃ [41920044-3; EMELCA Bioscience]) in round-bottom 96-well plates (650180; Greiner Bio-One) and incubated with indicated antibodies at 4°C for 45 min. After washing three times with FACS buffer, cells were incubated with 50 μl secondary antibody R-phycoerythrin (R-PE)–conjugated goat-antihuman IgG F(ab')₂ (diluted 1:500 in FACS buffer, 109-116-098; Jackson ImmunoResearch) for 45 min at 4°C protected from light. After washing twice, cells were resuspended in 150 μl FACS buffer supplemented with TO-PRO-3 iodine (1:650; Invitrogen by Thermo Fisher Scientific, T3605).

To assess binding to membrane-expressed 5T4, CHO-S cells (1 × 10⁶ cells/ml) were transfected with full-length human 5T4 (1.25 μg/ml). Transfection was performed using Freestyle Max transfection reagent (16447100; Thermo Fisher Scientific) and OptiPro SFM medium (12309050; Thermo Fisher Scientific), supplemented with 50 U/ml penicillin and 50 μg/ml streptomycin (Pen/Strep, DE17-603E; Lonza), according to the manufacturer's instructions. Cells were washed and resuspended in FACS buffer at 1 × 10⁶ cells/ml, seeded at 50,000 cells/well in round-bottom 96-well plates, and centrifuged at 300g, 4°C, for 3 min. After removal of supernatant, the cells were washed once in FACS buffer and incubated with 50 μl DuoBody-CD3x5T4 or bsIgG1-CD3xctrl at 4°C for 30 min. Cells were washed twice with FACS buffer and incubated with 50 μl secondary antibody R-phycoerythrin (R-PE)–conjugated goat-antihuman IgG F(ab')₂ (diluted 1:500 in FACS buffer, 109-116-098; Jackson ImmunoResearch) for 45 min at 4°C protected from light. After washing twice, cells were resuspended in 150 μl FACS buffer supplemented with TO-PRO-3 iodine (1:650).

Antibody binding to membrane-expressed CD3 or 5T4 was analyzed by flow cytometry on a BD LSRFortessa X-20 cell analyzer (BD Biosciences). Flow cytometry data were analyzed using FlowJo software. Binding curves were analyzed using nonlinear regression analysis (sigmoidal dose–response with variable slope) using GraphPad Prism software.

### Binding of DuoBody-CD3x5T4 to 5T4-expressing tumor cells and quantitative flow cytometry analysis

Tumor cells were washed after trypsinization (0.5% Trypsin–EDTA), seeded (30,000–50,000 cells/well) in round-bottom 96-well plates, and centrifuged. The tumor cells were washed in FACS buffer and

incubated with DuoBody-CD3x5T4 or bsIgG1-CD3xctrl at 4°C for 30 min. Cells were washed twice in FACS buffer and incubated with R-PE-goat-anti-human IgG F(ab')₂ (1:500 in FACS buffer) at 4°C protected from light for 30 min. Cells were analyzed on an iQue screener (Intellicyt Corporation). For quantitative flow cytometry analysis, a standard curve was generated in parallel using a Human IgG Calibrator Kit (CP010; Biocytex), essentially according to the manufacturer's instructions, which was used to interpolate the number of DuoBody-CD3x5T4 molecules bound per cell.

### T cell–mediated cytotoxicity, activation, and proliferation

Unless otherwise indicated, an optimized E:T ratio of 4:1 was used because an E:T ratio of 8:1 did not further reduce the IC₅₀ (see Fig S3B and C). Tumor cells were seeded (16,000 cells/well) in assay medium (Roswell Park Memorial Institute-1640 medium with 25 mM Hepes and L-glutamine [BE12-115F; Lonza] supplemented with 10% DBSI [20371-029; Life Technologies] and pen/strep [DE17-603E; Lonza]) in 96-well flat-bottom plates (655180; Greiner Bio-One) and left to adhere at 37°C, 5% CO₂ for 4 h. Next, effector cells and antibody, both diluted in assay medium, were added and incubated at 37°C for indicated time points. As a positive control for cytotoxicity, 16 μg/ml phenylarsine oxide (PAO; Sigma-Aldrich, P3075; dissolved in DMSO [D2438; Sigma-Aldrich]) was used. After 72 h, supernatant was transferred to round-bottom plates, and suspension cells were pelleted by centrifugation and used to assess T-cell activation by flow cytometry. Cell-free supernatants were used to measure cytokine, GZMB, and perforin levels. The remaining adherent cells were washed three times in PBS and incubated with 150 μl of a 10% Alamar Blue solution (DAL1100; Thermo Fisher Scientific; diluted in assay medium) at 37°C for 4 h. Alamar Blue absorbance was measured at 615 nm (OD₆₁₅) on an EnVision plate reader (PerkinElmer). The percentage of viable tumor cells (i.e., metabolically active cells) was calculated using the following equation: % survival = ([absorbance sample − absorbance PAO-treated target cells]/[absorbance untreated target cells − absorbance PAO-treated target cells]) × 100.

To assess T-cell activation, T cells were washed in FACS buffer and stained using the antibody panel described in Table S5 at 4°C for 30 min protected from light and analyzed on the BD LSRFortessa X-20 cell analyzer (BD Biosciences). Data were analyzed using FlowJo software.

To analyze T-cell proliferation, T cells were labeled with the CellTrace CFSE Cell Proliferation Kit (C34554; Life Technologies) according to manufacturer's protocol, before use in T cell–mediated cytotoxicity assays (E:T ratio = 8:1). CFSE dilution was measured by flow cytometry, and T-cell proliferation was determined using the proliferation modeling tool from FlowJo. Expansion index values were calculated according to the following equation: Expansion index = Total number of cells/number of cells at start of culture = $(G_0 + G_1 + G_2 + G_3 + G_4)/(G_0 + G_1:2 + G_2:4 + G_3:8 + G_4:16)$ with $G_n$ = number of cells in generation n peak (with n = 0–7).

Dose–response curves generated using four-parameter nonlinear regression analysis (GraphPad Prism) were used to determine at which concentrations DuoBody-CD3x5T4 induced half-maximal response (IC₅₀/EC₅₀ values).

## Cytokine, granzyme B, and perforin analysis

Cytokine production was measured in supernatants from the T cell–mediated cytotoxicity assays. A U-PLEX Proinflam Combo 1 (hu) multiplex assay (K15049K; MeSo Scale Discovery), custom-made combined U multiplex (MSD, K15067M-2) and R-plex assay (MIG [MSD, F210I-3] and GZMB [MSD, F213X-3]) and a Luminex assay (Milliplex MAP – human cytokine/TH17 panel [SPR1459; Millipore Sigma]) were used essentially according to manufacturer's instructions. Perforin and GZMB release were analyzed by ELISA using the human Perforin ELISA development kit (3465-1H-6; MabTech) and the human GZMB DuoSet ELISA kit (DY2906-5; R&D Systems), according to manufacturer's instructions. Absorbance was measured at 450 nm with an ELISA plate reader (ELx808 ELISA Reader; Biotek Instruments).

## Generation of CRISPR-Cas9 5T4, Fas, IFNGR1, 5T4/Fas, and 5T4/IFNGR1 KO MDA-MB-231 and Fas KO NCI-H1299 tumor cells

Virus production of the CRISPR-Cas9 constructs was performed in HEK293T cells (ATCC, clone CRL-11268). HEK293T cells were plated in CELLSTAR six-well plates (657160; Greiner Bio-One) at a density of $1 \times 10^6$ cells/well in DMEM with high glucose (BE12-709; Lonza) supplemented with 10% heat-inactivated DBSI, 200 mM L-glutamine, and pen/strep. The next day, HEK293T cells were transfected with a mixture of 1.2 $\mu$g sg_hTPBG_1 (MDA-MB-231), sg_hFAS_BB_1 (NCI-H1299), sg_hFAS_BB_2 (MDA-MB-231), sg_IFNGR1_2 (MDA-MB-231 single KO), and/or sg_IFNGR1_1 (MDA-MB-231, 5T4/IFNGR1 KO) in pRSGCCG-U6-CMV-Cas9-2A-Puro (Cat. no. 93484-15P, 93240-33P and 93240-34P, 95532-20, and 95532-19, respectively; Cellecta) with 1.2 $\mu$g of Ready-to-Use Lentiviral Packaging Plasmid Mix (CPCP-K2A; Cellecta) and 9 $\mu$l Lipofectamine 2000 (REF11668-027; Invitrogen) in 300 $\mu$l Opti-MEM/Glutamax (51985-026; Gibco), which was incubated at RT for 20 min before dropwise addition to the HEK293T cells. After 48 h, supernatant containing virus particles was collected from the HEK293T cells and filtered through a 0.2-$\mu$m filter (431219; Corning) and used to transduce MDA-MB-231 or NCI-H1299 cells supplemented with hexadimethrine bromide (final concentration: 8 $\mu$g/ml, 107689; Sigma-Aldrich). Transduced MDA-MB-231 or NCI-H1299 cells were selected by addition of 2 $\mu$g/ml puromycin (P9620; Sigma-Aldrich) for 9 d. Single-cell clones of the transduced MDA-MB-231 or NCI-H1299 cells were obtained by a limiting dilution. The KO of these genes was confirmed by flow cytometry after overnight incubation with 100 ng/ml IFN$\gamma$ (285-IF; R&D Systems) using the following antibodies: AF647-labeled IgG1-5T4-FEAR (produced in house), PE-labeled mouse antihuman Fas (MA1-19795; Invitrogen), PE-labeled mouse antihuman IFNGR1 (12-119942; Invitrogen), and BV711-labeled mouse antihuman PD-L1 (400354; BioLegend).

## Bystander kill mixing and transfer experiment

### Mixing experiment analyzed by flow cytometry

MDA-MB-231 parental cells were mixed with MDA-MB-231 5T4 KO cells in different ratios and seeded in 12-well plates. DuoBody-CD3x5T4 (0.1, 1, and 10 $\mu$g/ml) and effector cells (E:T = 4:1) were added. After 72 h of incubation, the supernatant (including, e.g., T

cells and dead tumor cells) was collected in a 15-ml centrifuge tube (188271; Greiner). The wells were washed twice with 0.5 ml PBS, after which adherent cells were detached with trypsin-EDTA. The wash steps and trypsinized cells were added to the 15-ml collection tubes. After centrifugation, the pellets were resuspended and stained with fixable viability stain 510 (FVS510, PBS [564406; BD BioSciences]) and EF450-antihuman CD3 antibody (48-0037-42; eBioSciences) and IgG1-5T4-FEAR-A647. After washing, the cells were resuspended in FACS buffer and measured by flow cytometry.

### Transfer experiment

MDA-MB-231 parental tumor cells and purified T cells were incubated with 10 $\mu$g/ml of DuoBody-CD3x5T4 or bsIgG1-CD3xctrl for 72 h. As a positive control for T-cell activation, T cells were stimulated with anti-CD3 (plates were pre-coated with 2 $\mu$g/ml anti-CD3 antibody [16-0037-85; eBioscience] at 37°C for 4 h) and anti-CD28 antibody (2 $\mu$g/ml, 16-0289-085; eBioscience) at 37°C for 72 h. Supernatant was transferred either directly (i.e., containing T cells and antibodies) or after pelleting of the T cells by centrifugation (i.e., cell-free supernatant) to MDA-MB-231 5T4 KO cells seeded in a 96-well flat-bottom plate 16 h prior. The MDA-MB-231 5T4 KO tumor cells were incubated with these supernatants or with freshly added T cells and antibody at 37°C for 72 h. After 72 h, supernatant and adherent cells (after trypsinization) were collected and analyzed for cytotoxicity (using the viability dye BV510 [564406; BD]) or Fas expression (FITC-labeled mouse antihuman Fas, 555673; BD Biosciences) by flow cytometry.

## In vivo mouse xenograft experiments

The CDX experiments were performed in compliance with the Dutch animal protection law (WoD) translated from the directives (2010/63/EU) and, if applicable, the Code of Practice "animal experiments for cancer research" (Inspection V&W, Zutphen, The Netherlands, 1999) and were approved by the Ethical Committee of Utrecht (project number AVD244002017-1264-01). The PDX experiments were performed in compliance with the US Department of Agriculture's Animal Welfare Act (9 CFR Parts 1, 2, and 3) as applicable. Tumor growth was evaluated twice per week using a caliper and tumor volumes (mm$^3$) were calculated from caliper measurements as 0.52 × (length) × (width)$^2$ (CDX) or 0.50 × (length) × (width)$^2$ (PDX). Individual mice were euthanized when the tumor volume exceeded 1,500 mm$^3$ or when the animals reached humane endpoints.

### Prostate CDX model in the PBMC IV model

Male NOD-$Prkdc^{em26Cd52}Il2rg^{em26Cd22}$/NjuCrl (NCG) mice (Beijing Vital Star Biotechnology) were inoculated SC with DU-145 cells ($5 \times 10^6$ cells in 100 $\mu$l PBS with Matrigel). When tumors reached a volume of 70–100 mm$^3$, mice were injected IV with $1 \times 10^7$ huPBMCs. In the first experiment, tumors were extracted 7, 14, and 21 d after PBMC administration, after which they were dissociated with the Tumor Dissociation Kit (130-095-929; Miltenyi Biotec) according to manufacturer's instructions and stained for human CD3 (Clone SK7, 564001; BD Biosciences) to determine human T-cell infiltration by flow cytometry. In the second experiment, mice were treated weekly with indicated dose levels of DuoBody-CD3x5T4 starting 7 d after PBMC administration.

### Breast cancer CDX model in humanized NSG-HIS mice

*Il2rg^tm1Wjl*/SzJ (NSG) mice (3–4 wk old) were injected in the tail vein with CD34$^+$ HSCs from healthy human donors, generating NSG mice with humanized immune systems (NSG-HIS mice, provided by the Jackson Laboratory). NSG-HIS mice were inoculated with MDA-MB-231 cells (5 × 10$^6$ cells in 100 $\mu$l PBS). Mice were randomized in treatment groups with equal distribution of tumor volume (100–150 mm$^3$), HSC donors, and percentage of human CD3$^+$ T cells within the human CD45$^+$ population. Treatment was performed IV twice weekly for 3 wk (2QW × 3). To analyze peripheral T-cell activation, blood samples were collected at 24 and 48 h (from the cheek pouch) and/ or at 72 h (terminal heart puncture) after treatment and collected in heparin tubes containing lithium-heparin (Microvette300LH, 20.1309; Sarstedt). Peripheral T-cell activation and cytokine levels were analyzed as described below (see the "Analysis of peripheral T-cell activation and cytokine levels in NSG-HIS mice" section). Mice were euthanized at 72 h after DuoBody-CD3x5T4 treatment. Tumors were excised and either dissociated for analysis of intratumoral T cells (flow cytometry, see Table S5) and cytokine production (Luminex) or formalin-fixed and embedded in paraffin for IHC and immunofluorescence (IF) analysis as described below (see the "IHC and IF analysis on breast cancer CDX tissue sections from NSG-HIS mice" section).

### Analysis of peripheral T-cell activation and cytokine levels in NSG-HIS mice

Blood cells were pelleted and stained with the antibody mixtures described in Table S5. After staining, cells were incubated for several min at RT in 150 $\mu$l red blood cell lysis buffer (10 mM potassium bicarbonate [P-9144; Sigma-Aldrich], 0.01 mM EDTA [ED-500G; Sigma-Aldrich], and 155 mM ammonium chloride [A-5666; Sigma-Aldrich] in water [0123; Aqua B Braun]). After washing, cells were analyzed at the BD LSRFortessa X-20 cell analyzer. To analyze peripheral cytokines, 50–100 $\mu$l blood was collected in EDTA tubes containing Di-Kalium-EDTA (Microvette300LH; Sarstedt, 16.444). Plasma was obtained by centrifugation at 300$g$ at RT for 10 min, and cytokines were quantified using multiplex analysis as described above (see the "Cytokine, granzyme B, and perforin analysis" section). MDA-MB-231 tumors were surgically resected immediately after euthanizing the animals. Half of the tumor was fixed in formalin and embedded in paraffin for IHC analysis. The other half was dissociated using the Tumor Dissociation Kit (130-095-929; Miltenyi Biotec) according to manufacturer's instructions. The remaining cell suspension was strained through a MACS SmartStrainer (70 $\mu$m), centrifuged at 300$g$ for 7 min, after which cells were resuspended in FACS buffer and stained using the antibody panel described in Table S5 at 4°C for 30 min protected from light. Cells were washed three times in FACS buffer, resuspended in 150 $\mu$l FACS buffer, and measured on the BD LSRFortessa cell analyzer (BD Biosciences). Data were analyzed using FlowJo software.

### IHC and IF analysis on breast cancer CDX tissue sections from NSG-HIS mice

Automated IHC stainings were performed on FFPE CDX tumor tissue slides using the Ventana Discovery autostainer platform. Singleplex

IHC stainings were performed with rabbit monoclonal antibodies against human CD3 (clone 2GV6, 790-4341; Roche), human CD8 (clone SP57, 790-4460; Roche), human CD25 (clone EPR6452, ab128955; Abcam), and human GZMB (clone EPR8260, ab134933; Abcam). Furthermore, a multiplex immunofluorescent (IF) assay was performed on Ventana Discovery to co-stain with CD3-FAM (Discovery-FAM kit, 7988150001; Roche), Ki-67-Rhodamine6G (Discovery-Rhodamine6G kit, 7988168001; Roche), and pan-cytokeratin-Cy5 (Discovery-Cy5 kit, 7551215001; Roche). IHC FFPE tissue sections (4 $\mu$m) were subjected to standard protocols for deparaffinization and CC1 antigen retrieval (Tris–EDTA buffer, pH 7.8, 950-124; Roche). For singleplex IHC, endogenous peroxidase activity was blocked (S2003; Dako Agilent), and primary antibody binding was detected using standard reagents (OmniMap anti-rabbit HRP [5269679001] and ChromoMap DAB [5266645001], Roche). Tissues were counterstained with hematoxylin (IHC) or DAPI (IF). CDX tissue sections were mounted in ClearVue (IHC, 4212; Thermo Fisher Scientific) or with ProLong Gold Antifade Mountant (IF, P10144; Thermo Fisher Scientific). Immunostained whole CDX tumor tissue slides were scanned with standardized scanning profiles on an Axio Scan Z1 (Zeiss). Cellular quantitation within viable intratumoral regions was performed on digital images with tailored image analysis algorithms using HALO software (Indica Labs). Cellular quantitation readouts were generated by calculating the percentage of marker-positive cells within the total number of nucleated cells within viable intratumoral areas.

### Prophylactic or therapeutic treatment with DuoBody-CD3x5T4 in tumor and PBMC co-engraftment models in NOD-SCID mice

A 1:1 mixture of huPBMCs (5 × 10$^6$ cells) and MDA-MB-231 cells (5 × 10$^6$ cells; in 200 $\mu$l PBS) was injected SC in the flank of female nonobese diabetic/severe combined immunodeficient NOD.C.B-17-Prkdc^scid/J (NOD-SCID) mice (obtained from Charles River Laboratories; age 7–8 wk) using a 29G needle (324892; BD). Treatment with indicated doses of DuoBody-CD3x5T4 (in 100 $\mu$l PBS, IV) or control (100 $\mu$l PBS) was initiated directly after inoculation of tumor cells and PBMCs (prophylactic) or after tumor establishment (therapeutic).

### Lung cancer PDX model

Humanized (tail vein injected with CD34$^+$ HSCs [derived from healthy human donors] at an age of 3–4 wk after sublethal irradiation) female NOD.Cg-Prkdcscid Il2rgtm1Sug/JicTac (NOG-HIS, Taconic Biosciences, Inc.) mice were used. LU7336 PDX–derived cells (EPO GmbH) were injected SC in the flank of the mice using a 29G needle (32892; BD). Mice were randomized in treatment groups with equal tumor volume distribution (100–150 mm$^3$), equal HSC donor distribution, and an equal percentage of human CD3$^+$ T cells within the human CD45$^+$ population. On treatment days, the mice were injected IV with the indicated doses of DuoBody-CD3x5T4 in 200 $\mu$l PBS or with 100 $\mu$l PBS as control. Treatment was performed 2QW × 3.

### Ex vivo patient tumor analysis

Dissociated tumor cells (DTCs) were procured from commercial vendors (Prisma Health/KIYATEC Biorepository or Discovery Life Sciences) who maintain strict ethical compliance, including fully

de-identified materials and stringent Institutional Review Board (IRB) and Ethics Committee compliance. Dissociated tumor cells were stained in KIYATEC FACS buffer (2% FBS [Corning] 2 mM EDTA [Invitrogen], in PBS [Fisher Bioreagents]) at $0.5 \times 10^6$ cells/ml with antibodies (Table S6) at 4°C for 10 min. After washing, viability marker DRAQ7 (BD Biosciences) was added at a 1:200 dilution before data acquisition on the CytoFLEX LX flow cytometer (Beckman Coulter). 3D patient-derived spheroid cultures were generated, and response testing was performed using KIYATEC proprietary technologies, including the KIYA-PREDICT (KIYATEC) testing platform (Shuford et al, 2019, 2021; Appleton et al, 2021; Reed et al, 2021). After 72 h of treatment, spheroid viability was assessed on a TECAN infinite M1000pro (Mannedorf) using a CellTiter-Glo 3D (Promega) assay. Viability was calculated as an average of seven replicates and normalized to untreated control. Spheroids were dissociated by incubation in ACCUTASE (STEMCELL Technologies) for 10 min at 37°C. T-cell activation was assessed by flow cytometry with an antibody panel (Table S6) as described above (see the "T cell–mediated cytotoxicity, activation, and proliferation" section).

## Statistical analysis and data presentation

Paired *t* tests, Mann–Whitney tests, or an ordinary one-way ANOVA with a posttest for linear trend were performed to assess differences between groups, as indicated in figure legends. For the in vivo studies, log-transformed tumor volumes were compared on the last day that all treatment groups were complete, that is, on the day of the first tumor-related death in each study, using an ordinary one-way ANOVA with a posttest for linear trend. Progression-free survival was analyzed using the Kaplan–Meier method and Mantel–Cox analysis with a Bonferroni correction for multiple comparisons. The relationship between $IC_{50}$ and 5T4 expression was assessed using nonparametric Spearman correlation analysis. In graphs with an offset y-axis, a dotted line is used to indicate $Y = 0$.

# Supplementary Information

# Acknowledgements

The authors would like to thank René Völker-van der Meijden for technical support on the biolayer interferometry (BLI) experiments; Patrick Franken for technical support on the 5T4 IHC analysis; Laura Smits-de Vries for technical support on the T cell–mediated cytotoxicity assays; Jeremy Gale for technical support on the Luminex experiments; Marcel Brandhorst for statistical support and technical support on the in vivo experiments; and Véronique Roolker-Knaup, Bart de Jong, and Jeroen van den Brakel for technical support on the in vivo experiments.

## Author Contributions

K Kemper: conceptualization, formal analysis, supervision, validation, investigation, visualization, methodology, and writing—original draft, review, and editing.

E Gielen: formal analysis, investigation, methodology, and writing—review and editing.

P Boross: formal analysis, supervision, investigation, methodology, and writing—review and editing.

M Houtkamp: formal analysis, supervision, investigation, visualization, methodology, and writing—review and editing.

TS Plantinga: formal analysis, supervision, investigation, visualization, methodology, and writing—review and editing.

SAH de Poot: visualization and writing—original draft, review, and editing.

SM Burm: visualization and writing—original draft, review, and editing.

ML Janmaat: visualization and writing—original draft, review, and editing.

LA Koopman: formal analysis, supervision, investigation, methodology, and writing—review and editing.

EN van den Brink: resources, methodology, and writing—review and editing.

R Rademaker: resources, methodology, and writing—review and editing.

D Verzijl: formal analysis, supervision, investigation, methodology, and writing—review and editing.

PJ Engelberts: formal analysis, supervision, investigation, methodology, and writing—review and editing.

D Satijn: conceptualization and writing—review and editing.

AK Sasser: conceptualization and writing—review and editing.

ECW Breij: conceptualization, supervision, and writing—review and editing.

## Conflict of Interest Statement

K Kemper, EN van den Brink, R Rademaker, D Verzijl, PJ Engelberts, and ECW Breij are inventors on granted or filed patents pertaining to DuoBody-CD3x5T4. All authors are employees and shareholders of Genmab. DuoBody-CD3x5T4 was co-developed by Genmab and AbbVie Inc (NCT04424641 [closed]).

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
