## [Reviewer comments · Life Science Alliance]

Life Science Alliance

Mechanistic and pharmacodynamic studies of DuoBody-CD3x5T4 in preclinical tumor models

Kristel Kemper, Ellis Gielen, Peter Boross, Mischa Houtkamp, Theo Plantinga, Stefanie de Poot, Saskia Burm, Maarten Janmaat, Louise Koopman, Edward van den Brink, Rik Rademaker, Dennis Verzijl, Patrick Engelberts, David Satijn, A. Kate Sasser, and Esther Breij

DOI: <https://doi.org/10.26508/lsa.202201481>

Corresponding author(s): *Esther Breij, Genmab; Kristel Kemper, Genmab*

Review Timeline:

Submission Date:	2022-04-13
Editorial Decision:	2022-06-27
Revision Received:	2022-08-22
Editorial Decision:	2022-08-22
Revision Received:	2022-08-23
Accepted:	2022-08-24

Scientific Editor: Novella Guidi

Transaction Report:

June 27, 2022

Re: Life Science Alliance manuscript #LSA-2022-01481-T

Dr. Esther C.W. Breij
Genmab
NETHERLANDS

Dear Dr. Breij,

Thank you for submitting your manuscript entitled "Mechanistic and pharmacodynamic studies of DuoBody-CD3x5T4 in preclinical tumor models" to Life Science Alliance. The manuscript was assessed by expert reviewers, whose comments are appended to this letter. We invite you to submit a revised manuscript addressing the Reviewer comments.

Thank you for this interesting contribution to Life Science Alliance. We are looking forward to receiving your revised manuscript.

Sincerely,

B. MANUSCRIPT ORGANIZATION AND FORMATTING:

Reviewer #1 (Comments to the Authors (Required)):

Kemper K. et al. describes a new bispecific antibody that recognizes both CD3 and the tumour antigen 5T4. This novel bsAb might enhance T cells responses against 5T4 bearing tumours. Authors show that the DuoBody-CD3x5T4 can activate T cells in the presence of 5T4+ cells, inducing their proliferation and tumour killing in vitro and in vivo. Since 5T4 antigen is expressed by most tumours, this bsAb could be an interesting therapeutic tool. Although the experimental approach is sound, the manuscript might benefit from several editing.

- Acronyms should be described the first time they are used.

The Materials and methods section is arduous to read. It is strongly recommended to be more concise and avoid duplicities. In addition, important information is not included:

- How was quantitative flow cytometry performed?
- PBMCs from blood donors were used; however, it is not clear whether the study has the approval by the appropriated ethic committee and whether the authors got donor's written consent. Please clarify this point. Also add the authorization code.
- In the following section: Generation of DuoBody-CD3x5T4 add the name of the two control molecules used in the manuscript.
- Authors state that in vivo mouse xenograft experiments were approved by the Ethical Committee of Utrecht. Adding the identification code is highly recommended.
- Statistical analysis. When more than two groups are analyzed, a correction by multiple comparison should be performed.

Results section.

- Pag 12, lines 7-8. Authors state that : "CRISPR-Cas9-mediated knockout of Fas did not affect 5T4 expression (Fig. 3A), and had no or minimal effect on activation of CD4+ and CD8+ T cells in the presence of DuoBody-CD3x5T4 (Fig 3B-C, and Supplementary Fig. 6A). While no activation is shown for CD4+ T cells, a significant difference was observed for CD8+ T cells. Therefore the use of "minimal effect" is not accurate.
- Pag 15, line 11. Why day 42 was chosen?
- Page 17 Line 10. Indicate which biopsies were positive.

Discussion:

While the DuoBody-CD3x5T4 induces a 100% killing of several cell lines such as MDA-MB-468 (Fig 1B) and RL95-2 (Fig 2B), in other cases, such as SW780 (Fig 2C) and , especially, human biopsies (Fig 7C), the killing is partial. These differences should be addressed in the discussion since they may impact on the therapeutic use and efficacy of this bsAb. In addition, the biological meaning of the bystander killing of 5T4 negative cells, and if that imply safety issues for clinical use should be addressed.

Figures.

All figure legends should clearly state the number of replicates, number of mice, number of human donors, etc. in each experiment. For instance, in Figure 1E, Is it just an experiment with samples from one individual?

Figure 2A. The color code is difficult to understand, since some of them are pretty similar (pancreatic vs lung; uterine vs prostate). The definition of the dotted line is missing.

Figure 3B. Correct the Y-axis legend: DuoBody-CD3x5T4.

Figure 4F. Describe the predicted black line. Define how it was calculated

Figure 5. Describe the meaning of the different symbols. In the figure legend clarify line 6: "...where all groups were complete."

Suppl Figure S2. Which concentration of the IgG1-5T4-FEAR were used. Please, describe the meaning of the dotted line. What is the meaning of gMFI?

Suppl Figure S4C. describe the control used. In this experiment, were tumor cell added? If so, please indicate.

Suppl Figure S8. Clarify line 8: "...where all groups were completed." Moreover, red triangles in F-J should be described in the figure legend.

Suppl Figure S8. Clarify lines 8: "...where all groups were completed." Moreover, red triangles in F-J should be described in the figure legend.

Reviewer #2 (Comments to the Authors (Required)):

In the manuscript by Esther Breij and co-workers a novel bispecific anti-CD3 x anti-5T4 bispecific antibody in the DuoBody format that crosslinks CD3e with the 5T4 tumor antigen is rigorously investigated in vitro and in various preclinical CDX and PDX tumor models.

It is convincingly demonstrated that the DuoBody-CD3x5T4 is able to mediate tumor cytotoxicity, T cell activation, proliferation and an enhanced production of various effector cytokines. Both CD45RA+ naïve as well CD45RO+ memory CD4/CD8 T cell were able exert tumor cell cytotoxicity in the presence of the DuoBody-CD3x5T4 albeit with slightly different kinetics. In addition to the classical granzyme B and perforin pathway, the authors describe a role for the Fas/CD95 death receptor pathway and/or the IFN-gamma signaling pathway. This is elegantly demonstrated by using of Fas and IFNGR1 knock-out tumor target cells. An unexpectedly high degree of bystander kill of 5T4-negative tumor cells is observed in cocultures with 5T4-silenced MDA-MB-231 cells. It is clearly shown that the bystander kill is partly due soluble factors secreted by activated T cells. Using 5T4/IFNGR1 double KO targets the effect could be attributed directly to IFN-gamma, while excluding a role of Fas for bystander killing. The authors demonstrate that autologous tumor-infiltrating lymphocytes can mediate kill of 5T4+ tumor targets in the presence of DuoBody-CD3x5T4 even when a PD-1 expression on TIL was pronounced. Altogether these are very interesting and profoundly supported findings advocating the potential use anti-5T4 x anti-CD3e bispecific antibodies for the treatment of solid tumors.

The highly incomplete maturation of enriched naïve T cells into CD45RO single+ memory cells remains an unexplained phenomenon. This could be related to the artificial DuoBody-mediated crosslinking of CD3e in the absence of costimulation, which could be tested by adding a costimulatory DuoBody. Hence it appears to be rather inappropriate to use a memory-like phenotype as a pharmacodynamics marker (page 18, line 21), and of course, there is no "actual memory formation" (page 18, line 23). This part of the discussion should be rephrased.

The finding that granzyme B secretion is more pronounced for CD4+ memory T cells and less for CD8+ memory T cells as compared with naïve counterparts (Fig. 1E, page 11, line 6) remains unexplained and should be discussed.

Page 10, lines 21-23: The description appears to be superficial. Only for CD4+ but not CD8+ memory T cells, this reviewer can confirm a more rapid upregulation of CD69 (Fig. 1C/D). The differences between naïve and memory T cells with regard to cytokine secretion at 24 h appear to be inconsistent.

Page 11, line 11: It should be made clear that the "memory phenotype" is a peculiar CD45RA+/CD45O+ population. Have the authors used the loss of CD62L expression to distinguish between effector memory and central memory T cells?

Reviewer #3 (Comments to the Authors (Required)):

In this paper Kemper et al. address in-depth mechanistic studies of a novel CD3 bispecific antibody (BsAb) in solid cancer. Naïve and memory CD4+ as well as CD8+ T cells mediated cytotoxicity in a FAS or IFNGR1 dependent fashion. In humanized xenograft models, antitumor activity of the BsAb was associated with intratumoral and peripheral blood T-cell activation. Additionally, the BsAb DuoBody-CD3x5T4 activated TIL from dissociated patient-derived tumor samples. Thus, preclinical mechanism of action of DuoBody-CD3x5T4 and its capacity to induce antitumor activity in preclinical models were studied in vitro, in vivo and ex vivo.

Several CD3 BsAbs have been developed for treatment of hematological malignancies, but the number of CD3 BsAb that are currently developed to target solid tumors limited. As such, this paper is of great interest, even though the described mechanisms are not novel.

Some minor questions remain, which could be easily addressed in a limited timeframe.

-The authors show that the different subsets are equally capable of mediating DuoBody-CD3x5T4 induced tumor cell kill, albeit with slightly different kinetics. Do they have an explanation why there is so little difference in killing capacity as it would be expected that CD8 cells are more cytotoxic than CD4 cells. And why is there a difference in activation and/or cytokine production? E.g. why is there so little granzyme production by memory CD8 cells? Did the authors check purity of isolated T cell subsets?

-In Figure 6E the authors show that T cells are present in the tumor after treatment with the control BsAb, but after therapy with DuoBody-CD3x5T4 cells are mostly located at the edge, resembling more an immune exclusion phenotype. Do the authors have an explanation? Did they also tested for cell death?

- Please explain the antigen 5T4 in the abstract, as it is not clear what the DuoBody CD3x5T4 targets.

- The authors state that: 'Flow cytometry analysis of the single cell suspensions demonstrated 5T4 expression on tumor cells of three out of four biopsies'. However, based on Figure 7C expression is minimal. As killing is observed in the presence of DuoBody-CD3x5T4, expression is functional. Nonetheless, the authors may consider nuancing the statement about the expression.

-It is not clearly described how the patient spheroids were formed.

First of all, we would like to thank the reviewers for their valuable feedback on our manuscript. We have done our utmost best to meet all comments and have made several changes in the manuscript. Please find below a point-by-point response to each critique.

Reviewer #1:

Kemper K. et al. describes a new bispecific antibody that recognizes both CD3 and the tumour antigen 5T4. This novel bsAb might enhance T cells responses against 5T4 bearing tumors. Authors show that the DuoBody-CD3x5T4 can activate T cells in the presence of 5T4+ cells, inducing their proliferation and tumour killing in vitro and in vivo. Since 5T4 antigen is expressed by most tumours, this bsAb could be an interesting therapeutic tool. Although the experimental approach is sound, the manuscript might benefit from several editing.

1. Acronyms should be described the first time they are used.

Thank you for bringing this to our attention, we have looked for all acronyms and defined them at first mention.

2. The Materials and methods section is arduous to read. It is strongly recommended to be more concise and avoid duplicities.

We acknowledge that the methods section is extensive. In the updated manuscript, we have shortened the text while still providing sufficient detail for any potential reader who would like to repeat our experiments. In addition, we have shortened the Supplementary Materials & Methods section and have merged sections that perhaps appeared to be duplications (eg, binding by BLI, binding by flow cytometry). In addition, we have removed the Study Design section.

3. In addition, important information is not included: how was quantitative flow cytometry performed?

The description of quantitative flow cytometry was originally in the Supplementary Methods under the header "Binding of DuoBody-CD3x5T4 to 5T4-expressing tumor cells". To clarify that this refers to the quantitative flow cytometry experiments, we have now added "and quantitative flow cytometry analysis" to the header and have moved this section to the main Methods.

4. PBMCs from blood donors were used; however, it is no clear whether the study has the approval by the appropriated ethic committee and whether the authors got donor's written consent. Please clarify this point. Also add the authorization code.

The PBMCs from blood donors are purified from buffy coats that have been commercially procured from a vendor (Sanquin) that collects blood for research purposes. Donors can object if they do not want a portion of their blood donation used for scientific research; if the entire blood donation is used for research, the donor's written consent is requested.

5. In the following section: Generation of DuoBody-CD3x5T4 add the name of the two control molecules used in the manuscript.

Thanks for pointing out this unclarity, we have added the names of the control antibodies in this section.

6. Authors state that in vivo mouse xenograft experiments were approved by the Ethical Committee of Utrecht. Adding the identification code is highly recommended.

The project number under which the in vivo studies were approved is AVD244002017-1264-01, which has been added to the materials and methods.

7. Statistical analysis. When more than two groups are analyzed, a correction by multiple comparison should be performed.

This is a very good point. Since we are interested in testing whether the observed effects are dose-dependent, we have reanalyzed experiments in which more than two groups were compared using an ordinary one-way ANOVA with a post test for linear trend. For the Mantel-Cox analyses, we have adjusted the p-values using the Bonferroni correction for multiple comparisons. We have adapted the Methods, Figure 5, Figure 6B-D, Figure S8, and Figure S9A-B, the corresponding figure legends, and the Results section accordingly.

8. Pag 12, lines 7-8. Authors state that: "CRISPR-Cas9-mediated knockout of Fas did not affect 5T4 expression (Fig. 3A), and had no or minimal effect on activation of CD4+ and CD8+ T cells in the presence of DuoBody-CD3x5T4 (Fig 3B-C, and Supplementary Fig. 6A). While no activation is shown for CD4+ T cells, a significant difference was observed for CD8+ T cells. Therefore the use of "minimal effect" is not accurate.

Indeed, we agree that "limited" might suggest that there was no effect; we have now changed the phrasing to "... and had no or only a small (albeit significant) effect on the frequency of activated CD4+ and CD8+ T cells in the presence of DuoBody-CD3x5T4..."

9. Pag 15, line 11. Why day 42 was chosen?

After Day 42, individual mice (particularly in the control group) had to be taken out of the study due to too large tumor volumes (a tumor volume cut-off 1,500 mm³ was used as a humane endpoint). This means that Day 42 was the last day that all treatment groups were still complete, i.e., until the first tumor-related death. This was mentioned in the figure legends, but we have now also added this explanation to the main text and to the methods for the purpose of clarity.

10. Page 17 Line 10. Indicate which biopsies were positive.

We added which biopsies were positive to this section: "Flow cytometry analysis of the single cell suspensions demonstrated 5T4 expression on tumor cells of three (OV1, UT1 and OV2) out of four biopsies".

11. While the DuoBody-CD3x5T4 induces a 100% killing of several cell lines such as MDA-MB-468 (Fig 1B) and RL95-2 (Fig 2B), in other cases, such as SW780 (Fig 2C) and, especially, human

biopsies (Fig 7C), the killing is partial. These differences should be addressed in the discussion since they may impact on the therapeutic use and efficacy of this bsAb.

This is a good point and something that we do not completely understand. We have now added the following paragraph to the discussion:

“Interestingly, while low levels of 5T4 expression are sufficient for complete tumor cell kill of certain tumor cell lines (e.g., the RL95-2 uterine cancer cell line; Fig. 2B) DuoBody-CD3x5T4 treatment of certain other tumor cell lines, including some cell lines with higher 5T4 expression levels (e.g., the SW780 bladder cancer cell line; Fig. 2C), resulted in only partial tumor cell kill. The reason for this partial resistance to DuoBody-CD3x5T4-induced tumor cell kill is not fully understood but might involve differential sensitivity of the tumor cells to Fas or IFN γ signaling. This could potentially also play a role in the partial kill observed in the 5T4+ dissociated patient-derived solid tumor samples ex vivo, although the latter could also be explained by differences in accessibility between 2D and 3D cultures, and a lower E:T ratio in the ex vivo experiments (Fig. 7E).”

12. In addition, the biological meaning of the bystander killing of 5T4 negative cells, and if that imply safety issues for clinical use should be addressed.

In this manuscript, we demonstrated that DuoBody-CD3x5T4 can induce bystander killing. We would like to emphasize that in our assays, bystander kill was dependent on the presence of 5T4+ on tumor cells in the same assay. Nevertheless, the clinical relevance of bystander killing, both with regard to safety and antitumor activity, is unknown. In our in vitro studies, bystander killing was evaluated in a 2D in vitro model, where IFN γ cannot diffuse and is accumulated in the well. We hypothesize that in a 3D tumor (model), the local concentration of IFN γ will be lower and therefore bystander killing may only occur in close proximity of tumor cells.

We hypothesize that eradication of the total tumor, including e.g. 5T4-negative endothelial cells or cancer-associated fibroblasts in close proximity to 5T4+ tumor cells will be beneficial for the antitumor activity. While target expression on at least a proportion of tumor cells is a key driver also for induction of bystander kill by a CD3 bsAb, we cannot exclude that bystander cytotoxicity may have an impact on the clinical safety of CD3 bispecific antibodies, including DuoBody-CD3x5T4.

In response to the reviewer’s comment, we have expanded on the potential clinical relevance of bystander kill difference between the in vitro and in vivo setting in the discussion: “The physiological relevance of DuoBody-CD3x5T4-induced bystander killing, its potential impact on safety and efficacy, and the role of IFN γ , IFNGR1 and Fas in this process will need to be explored in an in vivo setting, where local concentrations of IFN γ are expected to be lower due to diffusion.”

13. All figure legends should clearly state the number of replicates, number of mice, number of human donors, etc. in each experiment. For instance, in Figure 1E, Is it just an experiment with samples from one individual?

We have added the number of donors, replicates and mice used in each experiment in the figure legends.

14. The color code is difficult to understand, since some of them are pretty similar (pancreatic vs lung; uterine vs prostate). The definition of the dotted line is missing.

We can indeed see how it might be difficult to discern between some of the colors. We have therefore lightened the light shades and darkened the dark shades. In addition, we have removed the black strokes outlining the boxes. The dotted horizontal lines in Figure 2 (and in other figures) visualize the Y=0 line in graphs with an off-set Y-axis. To clarify this, we have added this explanation to the methods under the Statistical Analysis and Data Presentation header: "In graphs with an offset Y-axis, a dotted line is used to indicate Y=0."

15. Figure 3B. Correct the Y-axis legend: DuoBody-CD3x5T4.

Thank you for pointing this out, we have fixed the Y-axis title.

16. Figure 4F. Describe the predicted black line. Define how it was calculated.

The predicted black line in (F) refers to the outcome of the assay when no bystander killing is expected, e.g. if 50% of tumor cells are 5T4+, only 50% of tumor cells will be killed. This has been added to the Figure Legend.

17. Figure 5. Describe the meaning of the different symbols. In the figure legend clarify line 6: "...where all groups were complete."

The figure key explaining the meaning of the symbols is shown in the figure itself (on the right side of the figure, next to panels E and K). The statement "where all groups were complete" is now further clarified in the main text and the methods, as described above in response to a Question 9 from this reviewer.

18. Suppl Figure S2. Which concentration of the IgG1-5T4-FEAR were used. Please, describe the meaning of the dotted line. What is the meaning of gMFI?

The concentration used in Figure S2D of IgG1-5T4-FEAR is 30 µg/mL, which has been added to the Figure Legends. The dotted horizontal line visualizes the Y=0 line because this graph has an off-set Y-axis. As mentioned in the response to Question 14 of this reviewer, we have now added an explanation for the dotted line in the methods. gMFI is an acronym for geoMean Fluorescence Intensity; this abbreviation has now been explained in the Figure Legends as well.

19. Suppl Figure S4C. describe the control used. In this experiment, were tumor cell added? If so, please indicate.

In this experiment, purified T-cell subsets were incubated with tumor cells and DuoBody-CD3x5T4 (as described in Figure 1A) or a no antibody control. We added this information to the Suppl. Figure Legend of S4C.

20. Suppl Figure S8. Clarify line 8: "...where all groups were completed." Moreover, red triangles in F-J should be described in the figure legend.

A figure key explaining the colors and symbols (e.g., the red triangles) is included in Figure S8. The statement “where all groups were complete” is now further clarified in the main text and the methods, as described above in response to a previous comment from this reviewer.

Reviewer #2:

In the manuscript by Esther Breij and co-workers a novel bispecific anti-CD3 x anti-5T4 bispecific antibody in the DuoBody format that crosslinks CD3e with the 5T4 tumor antigen is rigorously investigated in vitro and in various preclinical CDX and PDX tumor models.

It is convincingly demonstrated that the DuoBody-CD3x5T4 is able to mediate tumor cytotoxicity, T cell activation, proliferation and an enhanced production of various effector cytokines. Both CD45RA+ naïve as well CD45RO+ memory CD4/CD8 T cell were able exert tumor cell cytotoxicity in the presence of the DuoBody-CD3x5T4 albeit with slightly different kinetics. In addition to the classical granzyme B and perforin pathway, the authors describe a role for the Fas/CD95 death receptor pathway and/or the IFN-gamma signaling pathway. This is elegantly demonstrated by using of Fas and IFNGR1 knock-out tumor target cells. An unexpectedly high degree of bystander kill of 5T4-negative tumor cells is observed in cocultures with 5T4-silenced MDA-MB-231 cells. It is clearly shown that the bystander kill is partly due soluble factors secreted by activated T cells. Using 5T4/IFNGR1 double KO targets the effect could be attributed directly to IFN-gamma, while excluding a role of Fas for bystander killing. The authors demonstrate that autologous tumor-infiltrating lymphocytes can mediate kill of 5T4+ tumor targets in the presence of DuoBody-CD3x5T4 even when a PD-1 expression on TIL was pronounced. Altogether these are very interesting and profoundly supported findings advocating the potential use anti-5T4 x anti-CD3e bispecific antibodies for the treatment of solid tumors.

1. The highly incomplete maturation of enriched naïve T cells into CD45RO single+ memory cells remains an unexplained phenomenon. This could be related to the artificial DuoBody-mediated crosslinking of CD3e in the absence of costimulation, which could be tested by adding a costimulatory DuoBody. Hence it appears to be rather inappropriate to use a memory-like phenotype as a pharmacodynamics marker (page 18, line 21), and of course, there is no "actual memory formation" (page 18, line 23). This part of the discussion should be rephrased.

We completely agree with the statement that CD3 bsAbs in general, including DuoBody-CD3x5T4, have not been shown to induce actual memory formation. The effect of CD3 bsAbs on T cells T-cell activation and downstream differentiation has been hypothesized to be independent of costimulation (e.g., Dreier et al., 2003), although the exact functional comparability with antigen-specifically stimulated T cells is unknown. Therefore, we hypothesize that an increase of memory-like T cells in response to crosslinking and T-cell activation could still serve as a PD marker as it will be a direct consequence of the biological activity of DuoBody-CD3x5T4, even if it is unclear if (and probably unlikely that) such differentiated cells can be functionally considered memory T cells”

We have slightly adapted the discussion to bring this point across more clearly: “An increased frequency of memory(-like) T cells in response to DuoBody-CD3x5T4 treatment might therefore serve as a PD marker. Of note, the observed in vitro T-cell differentiation is more likely to be a

consequence of artificial CD3 crosslinking by DuoBody-CD3x5T4 in the presence of 5T4+ tumor cells rather than actual antigen-specific memory formation”.

2. The finding that granzyme B secretion is more pronounced for CD4+ memory T cells and less for CD8+ memory T cells as compared with naïve counterparts (Fig. 1E, page 11, line 6) remains unexplained and should be discussed.

This is a valid point. The observed difference in extracellular GZMB for CD4 and CD8 memory T cells was consistent between 2 T-cell donors (see Figure below). Interestingly, a similar difference (more extracellular GZMB in cultures of activated CD4+ T cells than in cultures of activated CD8+ T cells) has been reported previously (Lin et al., 2014). We do not completely understand this phenomenon, but if we were to speculate, it could be due to a kinetic difference between CD4 and CD8 memory T cells, where GZMB levels might peak between 24-48 h for CD8 memory T cells and between 48-72 h for CD4 memory T cells. Conceptually, differences in the ratio of perforin to GZMB could also affect the amount of GZMB in the supernatant; the levels of perforin are known to be much lower in CD4+ T cells than in CD8+ T cells (Bade et al., 2005) while intracellular levels of GZMB are reported to be similar between subsets (Lin et al., 2014). One could speculate that lower perforin levels might hamper pore formation in target cells, resulting in less efficient diffusion of GZMB into the target cells and consequently in higher levels of extracellular GZMB.

To address this in the discussion, we have added the following sentence: “Interestingly, GZMB secretion was more pronounced for CD4+ than for CD8+ memory T cells. This is in line with a previous study, reporting slightly higher levels of extracellular GZMB secreted by activated memory CD4+ T cells compared to memory CD8+ T cells (Lin et al., 2014).”

Kinetics of granzyme B production by T-cell subsets after co-culture with breast cancer cells and DuoBody-CD3x5T4

The indicated T-cell subsets were incubated with MDA-MB-468 breast cancer cells (E:T ratio = 4:1, 2 donors) and increasing concentrations of DuoBody-CD3x5T4 for 24 h, 48 h, and 72 h. Supernatants of this co-culture were analyzed by a multiplex assay for the presence of granzyme B. Two donors (donor A [top row] and donor B [bottom row]) were tested in a T-cell mediated cytotoxicity experiment. A. Total CD3+ T cells. B. CD4+ naive T cells. C. CD4+ memory T cells. D. CD8+ naive T cells. E. CD8+ memory T cells.

3. Page 10, lines 21-23: The description appears to be superficial. Only for CD4+ but not CD8+ memory T cells, this reviewer can confirm a more rapid upregulation of CD69 (Fig. 1C/D). The

differences between naïve and memory T cells with regard to cytokine secretion at 24 h appear to be inconsistent.

Showing in Figure 1C-D, CD69 upregulation is highest on memory T cells (both CD4 and CD8 T cells) after 24 h when compared to naïve T cells. Indeed, CD69 does increase further on naïve CD8 T cells versus CD8 memory T cells; we hypothesized that the peak of T-cell activation might come earlier for memory than naïve CD8 T cells, probably between 24 and 48 h.

We agree that correlation between the cytokine production and T-cell mediated kill is not very clear and we removed this part in the text of the manuscript.

4. Page 11, line 11: It should be made clear that the "memory phenotype" is a peculiar CD45RA⁺/CD45O⁺ population. Have the authors used the loss of CD62L expression to distinguish between effector memory and central memory T cells?

We fully agree that the phenotype of CD45RA⁺/CD45O⁺ T cells is not clarifying what kind of T cells these are. Unfortunately, we did not include CD62L staining in this experiment to discriminate between effector memory and central memory T cells. This is mentioned in the Discussion: "...although it should be noted that memory T cells were evaluated without further distinction between effector and central memory subsets."

In addition, we have added the following to the Discussion:

"We hypothesized that these CD45RA⁺/CD45RO⁺ T cells are in the middle of their transition from a naïve phenotype (CD45RA⁺) to a more memory-like phenotype (CD45RO⁺). Longer incubation might have resulted in a full transition into a single CD45RO⁺ population."

Reviewer #3:

In this paper Kemper et al. address in-depth mechanistic studies of a novel CD3 bispecific antibody (BsAb) in solid cancer. Naïve and memory CD4⁺ as well as CD8⁺ T cells mediated cytotoxicity in a FAS or IFNGR1 dependent fashion. In humanized xenograft models, antitumor activity of the BsAb was associated with intratumoral and peripheral blood T-cell activation. Additionally, the BsAb DuoBody-CD3x5T4 activated TIL from dissociated patient-derived tumor samples. Thus, preclinical mechanism of action of DuoBody-CD3x5T4 and its capacity to induce antitumor activity in preclinical models were studied in vitro, in vivo and ex vivo.

Several CD3 BsAbs have been developed for treatment of hematological malignancies, but the number of CD3 BsAb that are currently developed to target solid tumors limited. As such, this paper is of great interest, even though the described mechanisms are not novel.

Some minor questions remain, which could be easily addressed in a limited timeframe.

1. The authors show that the different subsets are equally capable of mediating DuoBody-CD3x5T4 induced tumor cell kill, albeit with slightly different kinetics. Do they have an explanation why there is so little difference in killing capacity as it would be expected that CD8 cells are more cytotoxic than CD4 cells. And why is there a difference in activation and/or

cytokine production? E.g. why is there so little granzyme production by memory CD8 cells? Did the authors check purity of isolated T cell subsets?

Thank you for this elaborate question, we agree that at first glance, the relatively low levels of GZMB release by the memory CD8 T cells are puzzling. First of all, we have checked the purity of the populations after purification (shown in Suppl. Figure 4A), which is >75% for the memory CD8 T cells.

The lack of difference in killing capacity between CD4 and CD8 T cells could be explained by the MoA of CD3 bsAbs, which induce crosslinking of T cells and tumor cells, resulting in synapse formation irrespective of the recognition of peptides presented by MHC I or II complexes by the TCR. The differences in T-cell activation observed between the T-cell subsets could be caused by kinetic differences, as maximal T-cell activation might still occur between 24-48 h for memory T-cell subsets before going down again. The observed differences in cytokine production could be explained by the set-up of the experiment: as we have purified T-cell subsets, cytokines that would normally be used by other T-cell subsets will now accumulate in the well. The relatively low levels of GZMB release by the memory CD8 T cells also surprised us, but the phenomenon was consistent between 2 T-cell donors (see Fig in the response to Question 2 from Reviewer 2). Furthermore, similar results have been reported previously (Lin et al., 2014). We have also added the following sentence to the Discussion:

“Interestingly, GZMB secretion was more pronounced for CD4⁺ than for CD8⁺ memory T cells. This is in line with a previous study, reporting slightly higher levels of extracellular GZMB secreted by activated memory CD4⁺ T cells compared to memory CD8⁺ T cells (Lin et al., 2014).”

2. In Figure 6E the authors show that T cells are present in the tumor after treatment with the control BsAb, but after therapy with DuoBody-CD3x5T4 cells are mostly located at the edge, resembling more an immune exclusion phenotype. Do the authors have an explanation? Did they also tested for cell death?

We understand that by looking at the pictures, it might give the impression that these T cells are located at the rim of the tumor, however detailed evaluation confirmed the T cells are actually inside the tumor. If the T cells were excluded, they would be surrounding the tumor, but now they have started to infiltrate. To confirm this, spatial distribution analysis was performed for all tumors in this study, showing a random distribution throughout the tumor (see Figure below). This figure is now included in the manuscript as the new Figure S9E and referred to in the main text as follows:

“Although individual mice showed an increase in the frequency of CD3⁺ T cells and CD25⁺ cells 72 h after DuoBody-CD3x5T4 treatment, there was no statistically significant difference in the frequency of CD3⁺ or CD25⁺ cells (Fig. 6E and F), or in the spatial distribution of CD3⁺ cells within the tumors (Fig. S9E), between treatment and control groups.”

Necrotic areas were excluded from further analysis, and no staining for apoptotic markers (eg, cleaved caspase-3) was performed.

Distribution of CD3 staining in control and DuoBody-CD3x5T4 treated tumors

For assessment of the degree of intratumoral CD3+ T cell infiltration, digital spatial analysis was performed by HALO software (Indica Labs). For this, 10 bands were drawn on top of the tissue images and the number of CD3+ T cells per mm² of tissue surface was quantified per band.

3. Please explain the antigen 5T4 in the abstract, as it is not clear what the Duobody CD3x5T4 targets.

We have now adapted the abstract accordingly: “Crosslinking T cells with tumor cells expressing the oncofetal antigen 5T4 was required to induce cytotoxicity.”

4. The authors state that: 'Flow cytometry analysis of the single cell suspensions demonstrated 5T4 expression on tumor cells of three out of four biopsies'. However, based on Figure 7C expression is minimal. As killing is observed in the presence of DuoBody-CD3x5T4, expression is functional. Nonetheless, the authors may consider nuancing the statement about the expression.

We showed the histograms in Figure 7C to provide insight in the distribution of 5T4 over the cell population, however we acknowledge that due to the log-scale, differences in staining between the 5T4 and the control antibody are not immediately evident. Please find below the graphs representing the MFI of the isotype control and the 5T4 staining for each of the samples (note

that the different samples have been stained with different secondary antibodies, so MFI cannot be compared directly across different samples). Clear 5T4 staining is observed for OV1, UT1 and OV2. We have now added the MFI of the isotype control and 5T4 antibody to Figure 7C.

5. It is not clearly described how the patient spheroids were formed.

These experiments were performed by an external vendor (Kiyatech) and their techniques are proprietary. We have now indicated this more clearly in the Methods. More information about their 3D patient-derived spheroid cultures can be found in the following papers:

Reed et al., Cell 2021, A Functional Precision Medicine Pipeline Combines Comparative Transcriptomics and Tumor Organoid Modeling to Identify Bespoke Treatment Strategies for Glioblastoma
 Shuford et al., Neuro-Oncology Advances 2021, Prospective prediction of clinical drug response in high-grade gliomas using an ex vivo 3D cell culture assay
 Appleton et al., Cancer Immunology, Immunotherapy 2021, PD-1/PD-L1 checkpoint inhibitors in combination with olaparib display antitumor activity in ovarian cancer patient-derived three-dimensional spheroid cultures
 Shuford et al, Scientific reports 2019, Prospective Validation of an Ex Vivo, Patient-Derived 3D Spheroid Model for Response Predictions in Newly Diagnosed Ovarian Cancer

References:

Appleton et al., Cancer Immunology, Immunotherapy 2021, PD-1/PD-L1 checkpoint inhibitors in combination with olaparib display antitumor activity in ovarian cancer patient-derived three-dimensional spheroid cultures
 Bade et al., Int Immunol, 2005, Differential expression of the granzymes A, K and M and perforin in human peripheral blood lymphocytes
 Dreier et al, J Immunol, 2003, T cell costimulus-independent and very efficacious inhibition of tumor growth in mice bearing subcutaneous or leukemic human B cell lymphoma xenografts by a CD19-/CD3- bispecific single-chain antibody construct.
 Lin et al., BMC Immunol, 2014, Granzyme B secretion by human memory CD4 T cells is less strictly regulated compared to memory CD8 T cells
 Reed et al., Cell 2021, A Functional Precision Medicine Pipeline Combines Comparative Transcriptomics and Tumor Organoid Modeling to Identify Bespoke Treatment Strategies for Glioblastoma
 Shuford et al., Neuro-Oncology Advances 2021, Prospective prediction of clinical drug response in high-grade gliomas using an ex vivo 3D cell culture assay
 Shuford et al, Scientific reports 2019, Prospective Validation of an Ex Vivo, Patient-Derived 3D Spheroid Model for Response Predictions in Newly Diagnosed Ovarian Cancer

August 22, 2022

RE: Life Science Alliance Manuscript #LSA-2022-01481-TR

Dr. Esther C.W. Breij
Genmab
NETHERLANDS

Dear Dr. Breij,

Thank you for submitting your revised manuscript entitled "Mechanistic and pharmacodynamic studies of DuoBody-CD3x5T4 in preclinical tumor models". We would be happy to publish your paper in Life Science Alliance pending final revisions necessary to meet our formatting guidelines.

- please add your supplementary figure legends to the main manuscript text
- please add ORCID ID for both corresponding authors-you should have received instructions on how to do so
- please add the Twitter handle of your host institute/organization as well as your own or/and one of the authors in our system
- please incorporate the Supplemental Materials and Methods into the main Materials and Methods section. there is no size limit for this section
- the References mentioned in the Supplemental section and Supplemental Table S5 should be incorporated into the main Reference list, and removed from underneath Table S5 (they should remain within the table as listed)

Figure Check:

- the scale bars in Figure 6E are hard to see
- please be sure to mention panel F in the legend for Figure 7

A. FINAL FILES:

B. MANUSCRIPT ORGANIZATION AND FORMATTING:

Sincerely,

August 24, 2022

RE: Life Science Alliance Manuscript #LSA-2022-01481-TRR

Dr. Esther C.W. Breij
Genmab
NETHERLANDS

Dear Dr. Breij,

Thank you for submitting your Research Article entitled "Mechanistic and pharmacodynamic studies of DuoBody-CD3x5T4 in preclinical tumor models". It is a pleasure to let you know that your manuscript is now accepted for publication in Life Science Alliance. Congratulations on this interesting work.

DISTRIBUTION OF MATERIALS:

Again, congratulations on a very nice paper. I hope you found the review process to be constructive and are pleased with how the manuscript was handled editorially. We look forward to future exciting submissions from your lab.

Sincerely,
